# Feeding-Regime-Dependent Intestinal Response of Rainbow Trout after Administration of a Novel Probiotic Feed

**DOI:** 10.3390/ani13121892

**Published:** 2023-06-06

**Authors:** Marek Ratvaj, Ivana Cingeľová Maruščáková, Peter Popelka, Adriána Fečkaninová, Jana Koščová, Natália Chomová, Jan Mareš, Ondřej Malý, Rudolf Žitňan, Martin Faldyna, Dagmar Mudroňová

**Affiliations:** 1Department of Microbiology and Immunology, University of Veterinary Medicine and Pharmacy, 04181 Košice, Slovakia; marek.ratvaj@student.uvlf.sk (M.R.); ivana.cingelova@uvlf.sk (I.C.M.); jana.koscova@uvlf.sk (J.K.); natalia.chomova@student.uvlf.sk (N.C.); 2Department of Food Hygiene, Technology, and Safety, University of Veterinary Medicine and Pharmacy, 04181 Košice, Slovakia; peter.popelka@uvlf.sk; 3Department of Pharmaceutical Technology, Pharmacognosy and Botany, University of Veterinary Medicine and Pharmacy, 04181 Košice, Slovakia; adriana.feckaninova@uvlf.sk; 4Department of Zoology, Fisheries, Hydrobiology and Apiculture, Mendel University, 61300 Brno, Czech Republic; jan.mares@mendelu.cz (J.M.);; 5Research Institute for Animal Production Nitra, National Agricultural and Food Center, 95141 Lužianky, Slovakia; rudolf.zitnan@nppc.sk; 6Veterinary Research Institute, 62100 Brno, Czech Republic; martin.faldyna@vri.cz

**Keywords:** salmonid, probiotic feed, intestinal immune answer, gut microbiota

## Abstract

**Simple Summary:**

The use of probiotic bacteria in the aquaculture of salmonids, which are very sensitive to stress, represents a possibility to increase their resistance to diseases. We have developed a new, low-cost application form ensuring the rapid revitalization of probiotic bacteria. We tested continuous and cyclic feeding regimes with regard to their effect on the intestinal immune response and microbiota of rainbow trout. We found that continuous application stabilizes the intestinal microbiota in favor of beneficial lactic acid bacteria and does not cause unnecessary excessive stimulation of immunity, which can lead to an unwanted increase in energy demand. Cyclic application of probiotic feed has the same positive effect on the intestinal microbiota but provides the opportunity to stimulate the intestinal immune system of trout, for example, in periods of increased stress.

**Abstract:**

Intensive fish farming is associated with a high level of stress, causing immunosuppression. Immunomodulators of natural origin, such as probiotics or phytoadditives, represent a promising alternative for increasing the immune function of fish. In this study, we tested the autochthonous trout probiotic strain *L. plantarum* R2 in a newly developed, low-cost application form ensuring the rapid revitalization of bacteria. We tested continuous and cyclic feeding regimes with regard to their effect on the intestinal immune response and microbiota of rainbow trout. We found that during the continuous application of probiotic feed, the immune system adapts to the immunomodulator and there is no substantial stimulation of the intestinal immune response. During the cyclic treatment, after a 3-week break in probiotic feeding and the reintroduction of probiotics, there was a significant stimulation of the gene expression of molecules associated with both cellular and humoral immunity (CD8, TGF-β, IL8, TLR9), without affecting the gene expression for IL1 and TNF-α. We can conclude that, in aquaculture, this probiotic feed can be used with a continuous application, which does not cause excessive immunostimulation, or with a cyclic application, which provides the opportunity to stimulate the immunity of trout, for example, in periods of stress.

## 1. Introduction

Aquaculture is one of the fastest-growing branches of the economy, providing employment and food for millions of people worldwide, with production, consumption, and fish trade expected to increase steadily in the upcoming years [1]. Capacities to produce such quantities of fish are limited. Therefore, the concentration of fish on these farms increases. This leads to a higher level of stress that inevitably causes a risk of diseases and mortality with high economic significance. Such stress also causes worse reproduction and growth in animals [2,3,4]. Farm owners find themselves in a situation where they have to invest significant efforts and funds to prevent and control emerging diseases that can decimate fish monoculture, since the European Union banned the use of antibiotics as growth promoters in 2006 (European Commission in 2003). With stricter legislation to stop the development of antimicrobial resistance being enacted every year, and the focus on healthier foods that do not contain antimicrobial residues, there has been a need for alternative solutions that serve a similar purpose (e.g., probiotics, prebiotics, humic substances, phytoaditives, etc.).

A probiotic in aquaculture are defined as ‘a live, dead or component of a microbial cell that when administered via the feed or to the rearing water benefits the host by improving either disease resistance, health status, growth performance, feed utilization, stress response or general vigor, which is achieved at least in part via improving the host’s microbial balance or the microbial balance of the ambient environment’ [5]. In general, the most frequently used and studied are lactic-acid-producing bacteria (LAB), as reviewed by Chizhayeva et al. [6]. The interest in LAB is based on the fact that these bacteria are part of the natural gastrointestinal tract (GIT) microbiota and have the ability to tolerate the acidic environment and the action of bile acids in the digestive tract. By the fermentation of sugars, LAB produce lactic acid or other organic acids that lower the pH in the GIT and therefore create conditions that naturally protect the digestive tract from colonization by undesirable bacteria [7]. Other mechanisms by which probiotic LAB positively affect the intestinal microbiota include the production of inhibitory substances such as hydrogen peroxide or bacteriocins [8]. Probiotics affect the adherence of pathogens to the intestinal mucosa by blocking binding sites [9] and include other beneficial effects such as nutritional competition and immunomodulation [10]. The increasing economic importance of bacterial infections and stress in aquaculture has enhanced the interest in the study of defense mechanisms against these diseases. Teleosts, as well as warm-blooded animals, including humans, naturally possess components of innate and acquired immunity. Commensal gastrointestinal microbiota (CGIM) is the key to the optimal development of the immune system, and therefore, maintaining the physiological structure of CGIM is necessary to maintain the health of the host [11]. Currently, one of the most used methods for the regulation of the composition of CGIM is the application of probiotics and other bioactive compounds such as short-chain fatty acids [12], various plant extracts [13], or microalgae [14]. The immunomodulatory effect of probiotic bacteria has been confirmed in many studies by an observed increase in phagocytic activity, metabolic activity of phagocytes, complement activity, levels of lysozyme, the number of Ig+ cells in the gut, and the gene expression for the immunologically important molecules (e.g., cytokines), resulting in the improvement of the functions of the intestinal systems of fish as well as the resistance of fish to infections [6,15]. For these reasons, we consider the use of probiotics in aquaculture to be a potentially valuable tool going forward.

In our previous research, we isolated and identified two strains of autochthonous LAB from the intestine of healthy rainbow trout—*Lactobacillus plantarum* R2 (CCM 8674) and *Lactobacillus fermentum* R3 (CCM 8675), both with strong inhibitory activity against *Aeromonas salmonicida* and *Yersinia ruckeri*, which are currently among the most serious bacterial pathogens impacting salmonid fish. Both strains of LAB showed a high tolerance to the trout gut environment and the best growth abilities at different temperatures. Moreover, the strains complied with antibiotic susceptibility requirements, as regulated by the European Food Safety Authority. Both strains of LAB were sensitive to ampicillin, gentamicin, kanamycin, erythromycin, clindamycin, tetracycline, and chloramphenicol, and in addition, *L. fermentum* R3 was also sensitive to streptomycin [16,17]. *Lactobacillus plantarum* R2 also affected the cytokine response in the intestinal cell primary culture, which was challenged with *A. salmonicida* and *Y. ruckeri*. Strain R2 stimulated the immune response after *Y. ruckeri*-induced immunosuppression and the reduced inflammation caused by *A. salmonicida* [18].

A successful practical application of probiotics is highly dependent on the method of their application and the dosage form, as this is a relatively sensitive biological material. Choosing appropriate and effective dosage forms is necessary to obtain optimum probiotic effects. The ability of probiotic microorganisms to survive and multiply in the host strongly influences their positive effect [19]. Bacteria should be metabolically stable and active in the product and survive during the passage through the intestinal tract in large numbers. Under the directive of the FAO/WHO, the probiotic effects of food or feed may be declared only if they contain at least 10^6^ to 10^7^ live probiotic bacteria per gram. The current dosage forms for the large-scale production systems of animals are prepared for application to powdery and granular mixtures or water. Probiotics are incorporated in compound feed in various ways—during and after the processing of feed, by mixing in, or by spraying on the surface of the feed. The production of animal feed comprises granulation and compression, requiring high temperatures and pressures, which negatively affect the viability of the probiotic bacteria in the feed. Live probiotics stabilized by freeze-drying and mixed into powdery feed mixtures place high demands on storage conditions and demand often practically unreasonable demands for their activation. Fresh inoculants are unstable, and they have often limited, usually short shelf lives, which complicates storage; they are therefore are less useful in practice. For our probiotic strains, we have developed a new, cheap, and stable application form suitable for aquaculture. The mixture of biodegradable compounds together with microencapsulation proved to be effective in protecting the probiotic strain, which was applied directly to the surface of commercial pelleted fish feed. This technology, together with cold storage, ensured high concentrations of live probiotic bacteria for months after the preparation of this feed [20].

This technological procedure was used for the preparation of probiotic feed, the effect of which was subsequently monitored under in vivo conditions on rainbow trout. One of the experiment’s goals was to find a suitable application schedule (continuous versus cyclic application scheme) based on the evaluation of the influence on the immune response and intestinal microbiota. The effect on the immune response was assessed by monitoring the relative gene expression of immunologically important molecules in the trout intestines.

## 2. Materials and Methods

### 2.1. The Preparation of Probiotic Feed for Aquaculture

#### 2.1.1. Cultivation of Probiotic Bacteria and Preparation of Bacterial-Starch Hydrogel

The strain Lactobacillus plantarum R2 (CCM 8674) (according to the new taxonomy Lactiplantibacillus plantarum [21]) was used to prepare a probiotic diet. The strain was isolated from the intestinal content of rainbow trout (Oncorhynchus mykiss) reared at the fish farm Rybárstvo Požehy s.r.o. in the Slovak Republic [17]. An 18 h strain culture in 1 L MRS broth (HiMedia, Karnataka, India) was prepared for coating 1 kg of commercial pellets (EFICO Enviro 921 3 mm, BioMar, Brande, Denmark) at 37 °C in a shaker (PSU-20i Orbital Multi-Platform Shaker, Biosan, Riga, Latvia). After incubation, the bacterial culture in the MRS broth was centrifuged at 1000× *g* for 20 min at 22 °C (ROTINA 420R, Hettich, Tuttlingen, Germany). The cell pellets were washed twice in PBS (MP Biomedicals, Illkirch-Graffenstanden, France). The resulting cell pellet was considered a 100% cell suspension, from which a 25% suspension of L. plantarum R2 probiotic bacterial cells was prepared by resuspension in sterile saline. The dispersion was blended for 60 min on an electromagnetic plate (AccuPlate-Stirrer, Labnet International Inc., Edison, NJ, USA) with Starch 1500^®^ (Colorcon, Dartford, Kent, England) until a probiotic bacterial-starch hydrogel was formed.

#### 2.1.2. Coating of Commercial Aquafeed

The first coating layer on the commercial pellets (EFICO Enviro 921 3 mm, BioMar, Brande, Denmark) was created with colloidal silicon dioxide (Aerosil^®^ 200, Evonik Industries, Hanau, Germany) for 5 min at 40 rpm using a cube mixer (Erweka KB 20, Langen, Germany). Aerosil^®^ 200 adhered to the surface of the pellets formed a dry sorption layer for the next liquid layer. Then, the bacterial-starch hydrogel was evenly poured onto the aerosol-coated pellets and mixed again for 5 min at 40 rpm using a cube mixer (Erweka KB 20, Langen, Germany) (Table 1). The control diet was prepared via the same multiple-coating procedure, except that bacterial cells of *L. plantarum* R2 were added to the starch hydrogel layer. Finally, the pellets were dried at 35 °C for 4 h in a hot-air oven (Universal Oven UN 55, Memmert, Büchenbach, Germany).

The analytical constituents of both feeds are presented in Table 2. Feed analyses were performed in the laboratory of the Research Institute for Animal Production Nitra, National Agricultural and Food Center, Slovakia. The contents of dry matter, crude protein, fiber, fat, and ash were determined according to Commission Regulation (EC) No. 152/2009 [22]. Briefly, dry matter was determined by weighing and drying at 103 ± 2 °C. Nitrogenous compounds were measured according to the Kjeldahl method using the Kjeltec 8400 (FOSS, Hilleroed, Denmark) machine, fiber was determined via the Hennenberg-Stohman method using a Fibertec 8000 (FOSS, Hilleroed, Denmark), fat content was measured according to the Soxhlett-Henkel method using an SER 148 (Velp Scientifica Srl, Usmate Velate, Italy), and ash content was determined by weighing the sample and heating to 550 °C in a muffle furnace. Amino acids were measured by ion exchange chromatography on an automatic amino acid analyzer AAA 400 (Ingos, Prague, Czechia). Macro- and micronutrients were analyzed using an atomic absorption spectrometer, the iCE 3500Z (FOSS, Hilleroed, Denmark).

After drying the coated pellets, the total number of *L. plantarum* R2 was ~10^8^ CFU/g, as determined using the spread plating method on MRS agar (Merck, Darmstadt, Germany), anaerobically incubated (GasPak system, Becton Dickinson, San Diego, CA, USA) at 37 °C for 48 h. The probiotic diet was stored at 4 °C until it was fed to the experimental fish.

### 2.2. Experimental Animals and Sampling

For this experiment, 1000 juvenile rainbow trout (*Oncorhynchus mykiss*) with an average weight of 77.74 ± 0.38 g were divided into three groups based on their subsequent feeding regime, and each group was further separated and reared in three 1000 L tanks for a total of 9 tanks (3 tanks/group and 111 fish per tank) with continuous aeration and independent recirculation. Fish were maintained in a 12 h photoperiod (12 h light and 12 h dark). After a two-week adaptation period, during which the fish received the commercial feed (EFICO Enviro 921 3 mm, BioMar, Brande, Denmark) without the silicon-starch hydrogel coating, the fish started receiving probiotic feed or commercial feed with the silicon-starch hydrogel coating, respectively. The first group received probiotic feed continually (CON) during the whole duration of the experiment (11 weeks). The second group received the probiotic feed in cycles (CYC)—after 4 weeks of application, there was a 3-week break during which the fish received commercial feed coated with protective components and then were again fed with the probiotic feed for another 4 weeks. The third group served as a control group (CTRL) and received commercial feed coated with protective components. It was observed that the fish in each group consumed feed with a good appetite. The feed ratio was 2.68% of the fish’s body weight. Water temperature, pH, and oxygen saturation were monitored two times every day. The water properties during the experiment were as follows: temperature 17.5 ± 0.3 °C, oxygen saturation 89% ± 3%, and pH 7.1 ± 0.1.

Samples for relative gene expression analysis were taken 7 (1st sampling), 9 (2nd sampling), and 11 (3rd sampling) weeks after the start of the experiment, when 8 fish were randomly selected from each group. The fish were sacrificed by stunning with a blow to the back of the head, followed by a spinal cord transection. Each fish was dissected and samples of the middle intestine (the part immediately behind the pyloric intestine) were taken for further analysis. Samples were stored in RNAlater^TM^ solution (Invitrogen, Thermo Fisher Scientific Baltics, Vilnius, Lithuania) at −20 °C until processing. For better visualization of the sampling scheme, see Figure 1.

#### Ethical Statement

The experimental steps complied with national legislation—Act No. 246/1992 Coll., on the Protection of Animals against Cruelty, as amended, and Decree No. 419/2012 Coll., on the Protection, Breeding, and Use of Experimental Animals, as amended.

### 2.3. Measurement of Relative Gene Expression

Further processing of samples was conducted at the University of Veterinary Medicine and Pharmacy in Kosice. Samples were homogenized by a Precellys 24 Tissues Homogenizer (Bertin Technologies, Montigny-le-Bretonneux, France) using 6000 rpm in 2 × 30 s cycles with a 15 s break between them. RNA was isolated using an Omega E-Z Total RNA Kit (Omega Bio-tek, Norcross, GA, USA). The purity and concentration of RNA was measured using a spectrophotometer Nanodrop 8000 (Thermo Scientific, Waltham, MA, USA) at 260/280 nm. The isolated RNA was then transcribed into cDNA using a Quantitect Reverse Transcription Kit (Qiagen, Hilden, Germany). The PCR reaction was performed in 10 μL reactions, with each reaction consisting of 5 μL iQ™ SYBR^®^ Green Supermix (BioRad, Hercules, CA, USA), 0.5 μL of forward and 0.5 μL of rear primer (c = 10 μM/μL) for selected genes (Table 3), and 4 μL of cDNA with a concentration of 10 ng/μL. *β-actin* was used as a reference gene. All reactions were performed as triplicates and each assay contained a negative control without a cDNA template. Relative gene expression was analyzed in the samples of cDNA by qPCR using thermocycler iCycler CFX96 (BioRad, Hercules, CA, USA). The protocols for each gene are in Table 3. Relative gene expression was measured as ΔΔC_t_ (±SD) with CFX96 Manager Software (BioRad, Hercules, CA, USA).

### 2.4. Microbiological Screening

For microbiological screening, the determination of the counts of lactic acid bacteria, coliform bacteria, and total aerobic bacteria in the intestinal contents as well as adhered to the intestinal mucosa of trout was used. Samples were collected on the 4th (0th sampling), 7th (1st sampling), and 11th (3rd sampling) weeks. The middle part of the intestine was separated, and the content was gently squeezed into a sterile container. Subsequently, the intestine was rinsed three times with a sterile PBS and then cut longitudinally. The mucosa was scraped with the edge of a sterile glass slide. Feces and mucosal scraping samples were homogenized in isotonic saline solution (Stomacher, Seward, Worthing, UK), diluted by tenfold dilution, and bacterial counts were determined using the plate method. MRS agar plates (Merck, Darmstadt, Germany) incubated for 48 h at 37 °C under anaerobic conditions (GasPak system, Becton Dickinson, San Diego, CA, USA) were used to determine the amounts of LAB. Coliform bacteria were counted on MacConkey agar plates (HiMedia, Karnataka, India), and total aerobic bacteria were counted on blood agar plates composed of Columbia agar (HiMedia, Mumbai, India) and 5% of defibrinated sheep blood. The MacConkey and blood agar plates were incubated aerobically for 24 h at 37 °C. The bacterial counts are expressed in log_10_ of colony-forming units per gram of content/mucosa (log_10_ CFU/g) ± standard deviation.

### 2.5. Histo-Fluorescent In Situ Hybridization Analysis of L. plantarum in the Trout Intestines

The histo-FISH method was used to detect the location and abundance of *L. plantarum* in the trout’s middle gut. The samples were processed according to the HISTO-FISH protocol, which was standardized in our laboratory according to Madar et al. [29]. In brief, segments of the middle intestine, approximately 2–3 cm long, were ligated and collected without removing the intestinal contents. They were washed in PBS (pH 7.2) and immediately fixed for 4 h at 4 °C in Carnoy’s solution, composed of 60% ethanol, 30% chloroform, 10% glacial acetic acid, and 1 g of ferric chloride. After fixation, the samples were dehydrated through an alcohol series with ethanol concentrations of 70%, 90%, 96%, and 100%, embedded in paraffin, and subsequently, 10 µm thick sections were prepared. The sections were mounted on coated microscopic glass slides (Dako, Agilent Technologies, Glostrup, Denmark). In the next step, the slides were deparaffinized with xylene, rehydrated in a graded alcohol series from 100% to 50%, and washed in PBS. The slides of the highest quality were selected for processing using the above-mentioned standardized FISH procedure. For the detection of *Lactobacillus plantarum*, probe Lpla 990 5′ATCTCTTAGATTTGCATAGTATG3′, labeled with Texas red on the 5′ end with excitation 587 nm and emission 647–670 nm (red color), was used (Sigma Aldrich, St. Louis, MO, USA). The hybridization was carried out overnight at 52 °C in a dark humid atmosphere. After hybridization, the slides were stained with 4′,6-diamidino-2-phenylindole (DAPI) dye (excitation at 365 nm and emission at 445–450 nm) for 10 min. Finally, the slides were immersed in Vectashield mounting medium and covered by coverslips (Vector Laboratories, Burlingame, CA, USA). An epifluorescence microscope Carl Zeiss Axio Observer Z1 equipped with Software Carl Zeiss Zen 2 (ver. 3.3. blue edition) (Carl Zeiss, Oberkochen, Germany) was used for evaluation.

### 2.6. Morphometry of Intestine

During the second sampling, samples from the intestines of 6 fish per group were collected to assess the morphology of the intestine. Samples were washed in a physiological solution and fixed using 4% neutral formaldehyde. After fixation, dehydration was conducted using increasing alcohol concentrations (30%, 50%, 70%, 90%, and absolute ethanol). Samples were then clarified using benzene and embedded in paraffin. Histological sections of 5 µm (10 from each sample) were stained using hematoxylin–eosin. The height, width, and cross-section surface of the intestinal villi were measured using the Image C analytic system (Imtronic GmbH, Berlin, Nemecko) and the IMES program, using a color camera (SONY 3 CCD) and a light microscope (Axiolab, Carl Zeiss Jena, Nemecko) at 40× magnification.

### 2.7. Statistical Analysis

Statistical analysis was performed using GraphPad Prism Version 9.0.0 software (GraphPad Software, Inc., San Diego, CA, USA). Data normality was analyzed using the Shapiro–Wilk test for each individual parameter in each sampling. Since the data showed normality, they were further tested by one-way ANOVA followed by post hoc multi-comparison Tukey’s test to determine differences between the groups in each sampling. Differences between groups were considered significant if the *p*-value was lower than 0.05. Data are expressed as means ± standard deviations.

## 3. Results

### 3.1. Intestinal Immune Response

The level of relative gene expression of *igm* was measured as a marker of antibody response. In the first and the last sampling, no significant difference between groups was observed. In the second sampling, trout showed a significant increase in the intestinal *igm* gene expression compared to continually fed fish. (Figure 2).

The gene expression of *cd4* and *cd8* T cell co-receptors was studied to assess the cellular immune response at the gut level. The relative gene expression of *cd4* in the gut was significantly higher in the first sampling in fish continuously fed probiotics in comparison with the control as well as the cyclic group. During the second and the third sampling, there was no significant difference in *cd4* gene expression between the groups, but there appeared to be a trend of increasing expression in the CON and CYC treatments, respectively (Figure 3).

In the first sampling, the highest level of expression of *cd8* was recorded in the continuously fed group, with the cyclic group having a significantly lower level of expression in comparison to the continually fed group. On the contrary, in the second sampling, the cyclic group had a significantly higher expression when compared to the control and continual group, and in the third sampling, the values did not differ significantly in individual groups (Figure 4).

The cytokine response was assessed based on the gene expressions of pro-inflammatory cytokines (*il1-β* and *tnf-α*) and chemokine (*il8*). *tgf-β* was selected as a representative of anti-inflammatory cytokines.

The results show that *il1-β* relative gene expression was slightly, but not significantly, elevated in the continually fed group during the first and last sampling; however, such an increase was not observed during the second sampling. The cyclic group showed significantly lower levels during the first and third samplings when compared to the continual group, but these were not significantly different from the control (Figure 5).

The expression levels of *tnf-α* had the biggest variance among individual fish in their respective groups, contributing to a relatively high standard deviation. The cyclic group had a significantly lower level of *tnf-α* gene expression when compared to the continual group during the first sampling, but this was not significantly different from the control. During the second sampling, both experimental groups showed an increasing trend over the control, Lastly, during the third sampling, the expression levels of the *tnf-α* gene were approximately equal in both experimental groups and not significantly elevated above control levels (Figure 6).

The *il8* gene showed a significant decrease in expression in the experimental groups during the first sampling. During the second sampling, the expression levels of *il8* were significantly increased in the cyclical group when compared with the control. In the third sampling, both experimental groups had lower relative gene expression for this gene in comparison to the control, with the cyclic group’s expression being significantly lower than the control (Figure 7).

The relative gene expression of *tgf-β* did not differ between the groups in the first sampling. It was observed that the cyclic group had a higher level of expression in the second sampling after receiving probiotic feeding following the pause in feeding. *tgf-β* gene expression returned to the level of the control group after 2 weeks of receiving the probiotic feed, i.e., in the last sampling. The continually fed group did not show a significant change in the relative gene expression of *tgf-β* during the whole experiment (Figure 8).

*tlr9* belongs to a family of molecules that detect the presence of foreign cells in the intestine. During the first sampling, both experimental groups showed a significantly higher level of *tlr9* gene expression than the control group. During the second sampling the relative gene expression of *tlr9* was increased in the CYC group when compared to the control and continual group. During the last sampling, a significant increase in expression in the CON group over the control was observed, but the CYC group’s expression was not significantly different from that of the control (Figure 9).

### 3.2. Microbiological Results

To assess the impact of the new probiotic feed and feeding regime on the intestinal microbiota of trout, a screening of the representation of LAB, coliform bacteria, and total aerobes was carried out. The proportions of the monitored species of bacteria were very similar in the continuously and cyclically fed groups; no significant differences were recorded between these groups in any sampling. LAB counts were significantly higher in all three samplings in both probiotic-fed groups compared to the control (Figure 10). Conversely, the numbers of coliform and total aerobic bacteria were lower in the experimental groups than in the control group (Figure 11 and Figure 12). Before starting the application of probiotic feed, screening of LAB representation was carried out: the numbers were very low and ranged from 0–10^2^ CFU/g of mucosa or intestinal contents (data not shown). For this reason, the difference between the representation of LAB in the control and experimental groups is the most pronounced in the zeroth sampling, i.e., 4 weeks after the start of the administration of the probiotics.

### 3.3. Histo-FISH Analysis

For confirmation of the location and abundance of *L. plantarum* in the intestines of fish, histological FISH analysis was performed. A control in the form of a bacterial smear that was stained using the same method is presented in Figure 13J. No obvious changes in the morphology of intestines in either experimental group were observed (Figure 13D,G) when compared to the control (Figure 13A). No *L. plantarum* adhered to the intestinal wall was found in the control group (Figure 13B), and the concentration of bacteria observed in intestinal content was also low (Figure 13C). In the continually fed group, there were some *L. plantarum* present near the gut wall, although at a low concentration (Figure 13E). A high amount of *L. plantarum* was observed in the intestinal content of fish from this group (Figure 13F). Lastly, high amounts of *L. plantarum* and other bacteria were detected in the mucosa of the intestine in the cyclic group (Figure 13H), as well as in the gut content (Figure 13I). Overall, higher concentrations were observed in both experimental groups’ intestinal content compared to the control group.

### 3.4. Morphometry of the Intestine

In both probiotic groups, the surface of the villi was increased significantly compared to the control group. In the case of the cyclic group, a significant increase when compared to the continual group was observed. The height of the villi was not significantly different between the experimental groups and the control; however, the cyclic group also had a significantly increased height in comparison to the continual group. Lastly, the width of the intestinal villi was not significantly different between groups (Table 4). A microscopic view of the intestines is presented on Figure 14.

## 4. Discussion

LAB are part of the normal intestinal microbiota of fish [30]. The majority of successfully used probiotic preparations contain LAB, both in human and veterinary practice, including use in aquatic animals. When using autochthonous strains of probiotic microorganisms, there is a higher probability of their good adaptation to the conditions of the organism of the relevant animal species or its environment, as compared to non-autochthonous strains. Several experiments have been carried out in which the impact of the application of probiotics on the health and production parameters of trout was monitored. Some studies have used autochthonous strains [31,32,33], while others have used non-autochthonous ones [34,35,36,37], with varying results. In our experiment, we focused on testing the influence of the *L. plantarum* R2 strain, isolated from the intestine of healthy rainbow trout, which has demonstrated positive probiotic properties in previous in vitro [17,18] and in vivo experiments in Atlantic salmon [38]. In addition, we applied the strain using a newly developed, inexpensive application form, which is based on placing live probiotic bacteria in the protective starch hydrogel layer. This application form ensures the rapid revitalization of bacteria in the digestive tract and aquatic environment, as well as their long-term survival in storage conditions [20]. This new application form was used in an in vivo experiment for the first time. An important finding was that the trout willingly accepted it, so the food intake was not affected, as confirmed by fish weights in the experimental groups that did not differ or were even higher than in the control group (unpublished data). In addition, we tested two different feeding regimes—the continuous and intermittent application of probiotic feed to monitor the immune response of the fish. Often during the long-term application of immunomodulators, the immune system “adapts” and stops reacting to them. For this reason, we hypothesized that after a break in administration, cyclic application could activate stronger immunomodulation/immunostimulation than continuous application. Meanwhile, continuous application, which would be appropriate in farms with poor water quality, low zoohygienic standards, or higher levels of environmental risks, could cause unnecessary long-term immunostimulation, which is not desirable. Therefore, we focused primarily on monitoring the immune responses of fish, namely in the place of the direct action of the probiotic feed—in the trout intestine. We focused on selected markers of cellular as well as humoral immunity. Based on our previous results, we hypothesized that our probiotic strain could stimulate the expression of genes associated with non-specific immune response (TLR9, IL8) as well as for anti-inflammatory cytokine (TGF-β), but not for pro-inflammatory cytokines (IL1, TNFα). We also assumed there would be an influence on gene expression for some markers of cellular immunity. In order to examine the established hypotheses, in the cyclic group, samples were taken for immunological analyses after a 3-week break in the application of probiotics (first sampling); 2 weeks after reintroducing the probiotic feed (second sampling), where we expected stimulation; and after another 2 weeks (third sampling), where we expected adaptation to the long-term application of probiotics.

IgM is one of three classes of immunoglobulins that we can find in teleost fish, with the other two being IgD and IgT (IgT is sometimes called IgZ in certain teleosts) [39]. IgM is the most abundant Ig in teleost plasma, where it can be found in monomeric and tetrameric forms [40], and its concentration increases following bacterial infection, with a shift of isoform from monomer to tetramer [41]. Our results show that relative gene expression was elevated in fish from the CYC group after the reintroduction of probiotic bacteria into the intestinal tract when compared to continually fed fish. This difference between the CYC and CON groups was evident only 2 weeks after the re-introduction of the probiotic feed (second sampling); after another 2 weeks of probiotic feeding, there were no more differences between the groups (Figure 2). This difference between the groups could be explained by a slight decrease in the expression of the CON group during the second sampling, and a slight increase in that of the CYC group, even though the important takeaway is that none of the experimental groups differed significantly from the control. Gene expression in the continually fed group did not differ from the control throughout the experiment. This is in line with an experiment performed by Vazirzadeh et al. [42], who fed rainbow trout with three strains of probiotic bacteria, *L. buchneri*, *L. fermentum*, and *Saccharomyces cerevisiae*, and compared their influence on immune response after 30 and 130 days of feeding. After 30 days of application, they recorded non-significantly higher levels of total serum globulins in all three strains. After 130 days of application, globulin levels were lower compared to the previous sampling and did not differ from the control. An increase in total plasma immunoglobulins was also observed by Balcázar et al. [43] after the administration of probiotic LAB to brown trout. The highest levels were recorded after 3 weeks of *Lactococcus lactis* feeding. It was proven that fish have intestinal mechanisms to induce immune tolerance to nonpathogenic microorganisms [44] and this may also be the reason why we did not see a significant change in the continually fed group, as the fish organisms had adapted to the presence of probiotic in the intestine. After several days, the levels of the cyclical group decreased to the same levels as the control. It is important to note that expression in CYC-treated group also did not differ from the control during the whole experiment. One of the reasons for this may be that no immune response on the level of IgM is triggered by our probiotic once the organism receives it for a prolonged period, which can also be seen in the CON group. Another reason may be that although the expression has not changed, the serum level of IgM may be increased, as observed by the studies mentioned above, and therefore, a change in expression is not triggered.

Molecule CD4 is found on the surface of Th lymphocytes, where it functions as a coreceptor in the process of the recognition of antigens. CD4+ T cells are responsible, then, for the production of cytokines and thus are essential in the regulation of immune response [45]. Studies on different fish species with different probiotics showed only insignificant changes in gene expression for *cd4* in the intestine of experimental animals [46]. A significant increase in the number of CD4 cells at the intestinal level was recorded after infection with viral hemorrhagic septicemia virus in rainbow trout [47]. Our results show an increase in gene expression for *cd4* in the CON group as compared to the control as well as the CYC group, but only during the first sampling. In the following samplings, the level of *cd4* gene expression in the groups did not differ, and surprisingly, even in the second sampling in the CYC group, there was no significant increase: only a weak tendency to increase was recorded (Figure 3). This difference from the work of Picchietti et al. [46] could be the result of the use of an autochthonous strain of probiotics. The increase in relative expression could potentially prove beneficial to the host organism since its response to pathogens could be more rapid, while the level of expression of *cd4* was not high enough to cause any detrimental effect on the organism as a whole. This hypothesis would need to be proven by further research.

CD8 can be found as a co-receptor on the surface of cytotoxic T-lymphocytes. They are predominantly found in the thymus, gill, and intestine, with low concentrations in the blood, spleen, and pronephros of fish [48]. The function of these cells in teleosts is similar to that of these cells in mammals, which is a defense against virus-infected cells [49] and foreign cells [50]. Apart from cytotoxic T lymphocytes, there is also a population of dendritic-like cells, that have CD8 molecules, and their role is thought to be in the regulation of local mucosal immunity and immune tolerance in teleosts [51]. This function is confirmed in mammals [52]. Since our measurement of relative gene expression is not specific to a certain population of CD8 cells, it identifies all subpopulations of CD8+ cells of fish including dendritic-like cells. As observed in our experiment, the relative gene expression for this molecule was highly elevated after the fish in the cyclic group started receiving the probiotic feed again (Figure 4), indicating the immunomodulatory potential of this feeding regime. As we assumed, this stimulation is only temporary, and in the third sampling, when fish in both the CYC and CON groups received probiotic feed for 4 weeks, there were no significant differences between the groups. The significant downregulation in the CYC group compared to the CON group during the first sampling may be the result of a feeding break, as at this point the fish in the CYC group had not received the probiotic feed for 3 weeks. Importantly, however, there were no significant differences between the experimental groups and the control in this sampling. The levels of the continually fed group, which across the whole experiment did not significantly exceed those of the control group, further suggest some sort of stabilization of the CD8+ cell population in the intestine, although as stated before, it is not clear what proportion of the CD8+ cell population is formed by DC-like cells.

Interleukin 1 is considered to be one of the most important pro-inflammatory cytokines in fish [53]. During the experiment, measured levels of gene expression were not significantly different between the CON or CYC groups and the control group (Figure 5). After a prolonged period of consuming probiotic strains (21 and 28 days), similar results were recorded in grass carps, where levels of *il1-β* in the experimental group did not exceed the control [54], although in the mentioned work there was significant upregulation of *il1-β* expression after 7 days of receiving the probiotic feed. We did not observe such a phenomenon in the cyclically fed group after these fish had started receiving probiotics after a pause. Interestingly, the gene expression of *il1-β* was, in our experiment, significantly lowered in the cyclic group when compared to the continual group (Figure 4). This difference was, however, observed after the pause in probiotic feeding of the CYC group and during the third sampling, when both groups were receiving the same feed. The statistically significant difference between the experimental groups appears to be primarily due to the slight increase in values in the CON group. It is also worth noting that there has been research that observed lowered levels of this cytokine in fish fed probiotics [46], and also that these levels were increased [55] even after several weeks, further proving that interaction between different probiotic bacteria and different species of fish and conditions yields specific results. Overall, the lack of an increase in the expression of *il1-β* could mean that our probiotic has not triggered an inflammatory response at the intestinal level, which would be considered unwanted stress for the fish organism.

TNF-α is a pro-inflammatory cytokine that has many immunological functions, which are aimed at the regulation of cellular response as well as the modulation of other cytokines. The *tnf-α* gene is commonly expressed in many tissues of rainbow trout after stimulations with pathogens, their components, prebiotics [56], and also vaccines [57]. Similar to the results of the *il1* gene, the gene expression for TNF-α in the experimental groups did not differ from the control (Figure 6), and we also noted a significantly lower expression of the *tnf-α* gene in fish that did not receive probiotics for 3 weeks (CYC group) compared to the continuously fed group. This is in contrast to the work of Wu et al. [53], who administered probiotics based on autochthonous strains of *Shewanella xiamenensis* to grass carp and recorded a long-term significant increase in the relative expression of the *tnf-α* gene compared to the control, lasting from the seventh to the twenty-eighth day after the start of the application. Increased levels of gene expression for *tnf-α* in fish post probiotic feeding have also been demonstrated by other works [33,58]. However, it is important to note that the levels of *tnf-α* in the mentioned works were highest in the days immediately after feeding experimental animals with probiotics. As with *il1-β*, the lack of a significant increase in the levels of the experimental groups in comparison to CTRL suggests no negative stimulation of the immune system towards inflammation in healthy fish.

Interleukin 8 is a typical pro-inflammatory cytokine with chemotactic properties that responds very quickly to antigenic stimulation. The observed increase in relative expression in the CYC group after those fish started receiving probiotics again (Figure 7) correlates with the results of Wu et al. [54], where a sharp increase in the expression of *il8* over that of the control was recorded within a week of feeding fish probiotics. In the same research, it was shown that after 14 days, the levels of *il8* relative expression remained just slightly increased in the experimental groups when compared to the control. An increase in *il8* mRNA abundance in the intestine of fish fed probiotics after 21 days was also previously recorded [33]. On the contrary, when the CYC group did not receive the probiotic feed for 3 weeks, the expression level of the *il8* gene was reduced below the control level. This confirms the rapid stimulation of IL8 production and also the rapid stabilization in the absence of antigenic stimulation. We observed a decrease in gene expression for IL8 below the levels of the control group in the CON group after 7 weeks of consuming our probiotic feed, but no differences in the following two samplings. We suppose that in the fish fed continuously, the immune system adapted to the presence of beneficial bacteria, and the expression of this gene was stabilized. Finally, it was also confirmed in an in vitro experiment on trout cells that the effect of probiotics on the expression of the *il8* gene is strain-dependent. While some strains of probiotic bacteria increase expression, others decrease it. However, the most significant changes were induced by pathogenic bacteria [23].

TGF-β is a cytokine that possesses many regulatory functions. It is responsible for the suppression of the immune system to protect the organism against autoimmune diseases and maintains immune tolerance [59]. Its regulatory function, which is parallel to that in the mammalian immune system, was also found in teleosts [60]. It was observed that the relative gene expression of *tgf-β* increases after infection of the internal organs of fish, including the intestine [61]. As with the relative expression of previously mentioned molecules, an increase in the expression of *tgf-β* could be seen in the cyclically fed group after the reintroduction of probiotic feed, with a decrease over time (Figure 8). Probiotic-bacteria-fed Nile tilapia also showed an increase in the relative gene expression of *tgf-β* [62]. It is also important to note that the expression of *tgf-β* did not decrease significantly below the levels of the control group after the fish received our feed; in the CON group, the level of expression was the same as in the control group throughout the experiment. A decrease was observed in Atlantic salmon following a soybean meal diet that promoted proinflammatory reactions in the intestine [63]. This result in the CON group would suggest that our diet is safe even after prolonged consumption and does not weaken the immune system. We hypothesize that since no increase in the gene expression for TGF-β was observed in the CON group, continuous treatment with probiotics may not be as effective in treating intestinal inflammation as cycle feeding. However, this hypothesis will have to be confirmed in further experiments. Furthermore, an increase in TGF-β can have a therapeutic effect during intestinal inflammation [64]; therefore, the use of our probiotic feed could also prove effective in the treatment of such inflammation based on stimulating effect observed in the CYC group. Moreover, the anti-inflammatory potential of our R2 strain was confirmed in Atlantic salmon with enteritis induced by a pro-inflammatory soybean meal-based feed [37].

The innate part of the immune system recognizes the molecular structures of microorganisms. These pathogen-associated molecular patterns (PAMPs) are highly conserved and specific to certain microorganisms. The host organisms have developed a set of receptors to recognize PAMPs, one of them being TLR9 [65]. TLR9 in teleosts recognizes and binds to procaryotic (or viral) DNA after it is internalized in endosomes of mononuclear phagocytes [66] but can be also located on the surface of intestinal epithelial cells [67]. It has been suggested that TLR9 is stimulated by probiotics or their DNA [68], and our work shows similar results, with a higher increase in the gene expression of *tlr9* in the CYC group after the reintroduction of probiotics in the feed (Figure 9). Elevation of *tlr9* expression was also observed during the first sampling in both experimental groups, and in the CON group during the last sampling, when the level of expression in the CYC group was equal to the control. These increased levels of *tlr9* may have activated the innate immune system of the host, providing better preparedness for potential infection. As for the reason behind the overall increase in *tlr9* expression in both experimental groups across the whole experiment, there is a possibility that no tolerance is established since TLRs are a part of an innate immune response that triggers rapidly and has no memory function. Therefore, continuous exposure to bacteria stimulates the continual expression of this gene. However, this does not explain the low levels of expression in the CON group in the second sampling and in the CYC group in the third sampling, and further study is needed for a complete understanding of TLR–probiotic interaction.

One of the most important effects of probiotic preparations lies in their positive influence on the intestinal microbiome. In our previous experiment on Atlantic salmon, to which we administered probiotic strains *L. plantarum* R2 or *L. fermentum* R3 sprayed onto the surface of feed granules using a high-pressure coulter, intestinal bacterial communities were analyzed via high-throughput 16S rRNA gene amplicon sequencing. We found that the diversity of microbiota adhered to the intestinal mucosa was significantly increased. Both of the used strains significantly dominated in the intestinal contents as well as on the surface of the mucosa [69]. Similarly, in the present study, after the administration of *L. plantarum* R2 to trout, high numbers of LAB were recorded in the intestinal contents as well as on the mucosa of the middle intestine (Figure 10). On the contrary, a significantly lower representation of enterobacteria (Figure 11) and total aerobic bacteria (Figure 12) was observed compared to the control. As shown by the results of the HISTO-FISH analysis (Figure 13), *L. plantarum* dominated both in the digesta and on the mucosal surface. No significant differences in the representation of the observed bacterial species were found between the continuously and cyclically fed groups. Interestingly, even after a 3-week break in the feeding of probiotic feed, there were no differences in the amounts of LAB in the digesta or on the intestinal mucosa between the cyclic and continuous groups. In the salmon experiment, after the end of the application, the amounts of *L. plantarum* R2 and *L. fermentum* R3 slowly decreased, and after 14–20 days, they reached only 10^1^–10^2^ CFU/g (unpublished data). This means that the R2strain better colonized the gut of trout as compared to salmon, which confirms the better adaptation of the strain to the conditions of the digestive tract of the trout, from which it was originally isolated, as well as to fresh-water conditions. Although many enterobacteria are symbiotic, many are pathogenic, or potentially pathogenic, bacteria. Therefore, the ratio between LAB and enterobacteria is often considered as an indicator of the health of the intestinal microbiota. The shift in the favor of LAB indicates a positive effect of *L. plantarum* R2 on the intestinal microbiota.

In both probiotic groups, the surface of the villi was increased significantly compared to the control group (Table 4), which was caused by the density of the villi. Similar results where the application of commercial probiotics increased the density of villi in the intestine have been observed in poultry [70]. Such an increase allows for better absorption of nutrients, thus providing a better growth rate and conversion rate. The height of the villi was not significantly different between the experimental groups and the control. In a different study, it was observed that probiotic bacteria *Lactobacillus plantarum* and *Lactobacillus rhamnosus* caused an increase in the height of the villi in common carp [71], suggesting that not all probiotics have an equal effect on the target organism. Lastly, the width of the intestinal villi was not significantly different between groups, and the same result was recorded in a different study using multiple species probiotic preparation in pigs [72].

## 5. Conclusions

In conclusion, autochthonous trout probiotic strain *L. plantarum* R2 showed very good colonization abilities with a positive effect on the intestinal microbiota. LAB dominated in the intestinal contents as well as on the mucosa of the middle intestine of trout, even 3 weeks after the end of probiotic feed application. By comparing the two feeding regimens, we found that during the continuous application of probiotic feed, the immune system adapts to the immunomodulator, and most of the tested cytokines, including pro-inflammatory cytokines, did not significantly stimulate the intestinal immune response. The exception was the increased expression of the tlr9 and cd4 genes. On the other hand, after a 3-week break in probiotic feeding and subsequent reintroduction of probiotics, there was a significant temporary stimulation of the gene expression of molecules associated with both cellular and humoral immunity (*cd8*, *tgf-β*, *il8*, *tlr9*), without affecting the relative expression of the *igm* gene or the pro-inflammatory *il1* and *tnf-α* genes. Moreover, the probiotic feed significantly increased the absorption area of the intestine (Figure 15). Based on the obtained results, we can conclude that continuous application can be used for stabilization of gut microbiota in favor of LAB without overstimulation of intestinal immunity. However, we prefer the cyclical application, which provides the opportunity to modulate the immune response in critical periods associated with stress, such as transport, change in feed, bad weather conditions, etc. In addition, the newly developed application form proved to be suitable, as it ensures the long-term survival of the probiotic strain, is cheap with a simple preparation technology, and is accepted by the fish.

## Figures and Tables

**Figure 1 animals-13-01892-f001:**
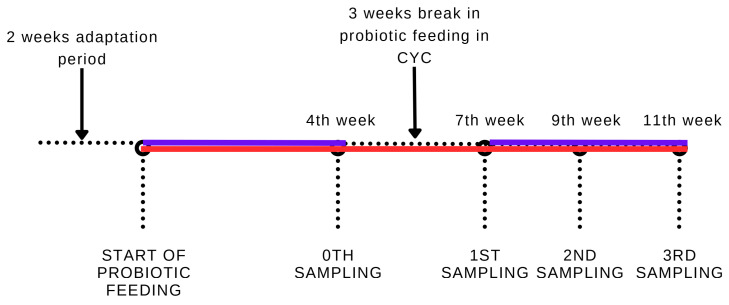
Scheme of application and sampling of rainbow trout fed probiotic feed. The red line group received probiotic feed continually (CON); the blue line group received probiotic feed cyclically (CYC).

**Figure 2 animals-13-01892-f002:**
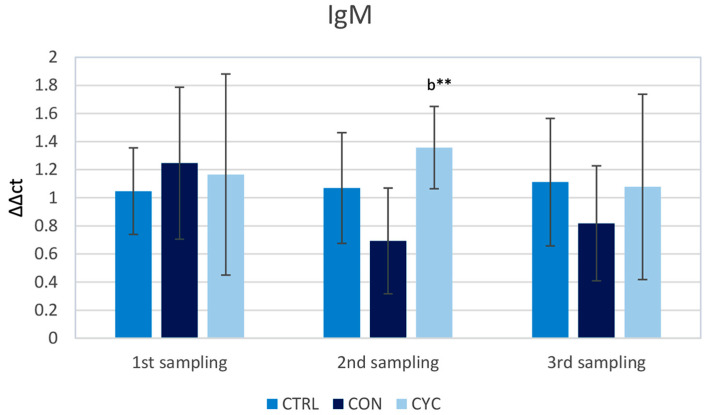
Influence of continuous and cyclic application of probiotic feed on the relative *igm* gene expression in the trout gut. b—significantly different from group CON; ** *p* < 0.01.

**Figure 3 animals-13-01892-f003:**
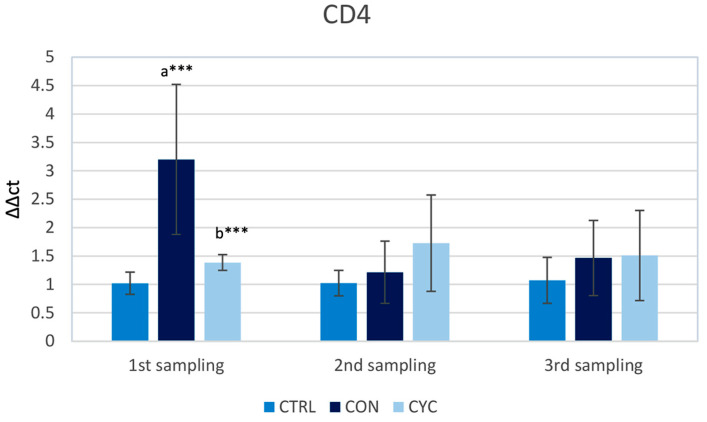
Influence of continuous and cyclic application of probiotic feed on the relative gene expression of *cd4* gene in the trout gut. a—significantly different from the control group; b—significantly different from group CON; *** *p* < 0.001.

**Figure 4 animals-13-01892-f004:**
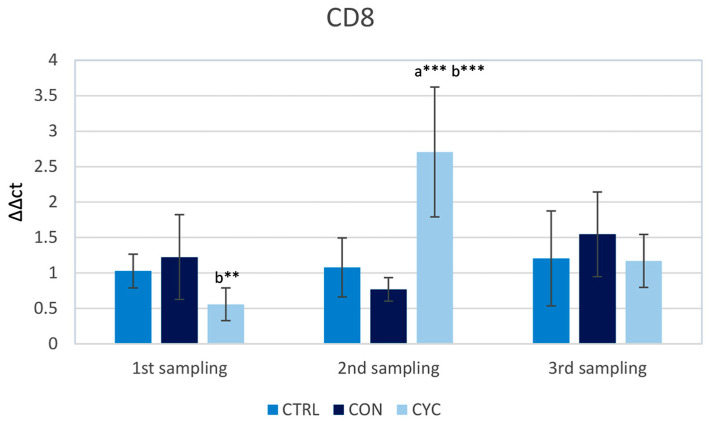
Influence of continuous and cyclic application of probiotic feed on the relative gene expression of *cd8* gene in the trout gut. a—significantly different from the control group; b—significantly different from group CON; ** *p* < 0.01, *** *p* < 0.001.

**Figure 5 animals-13-01892-f005:**
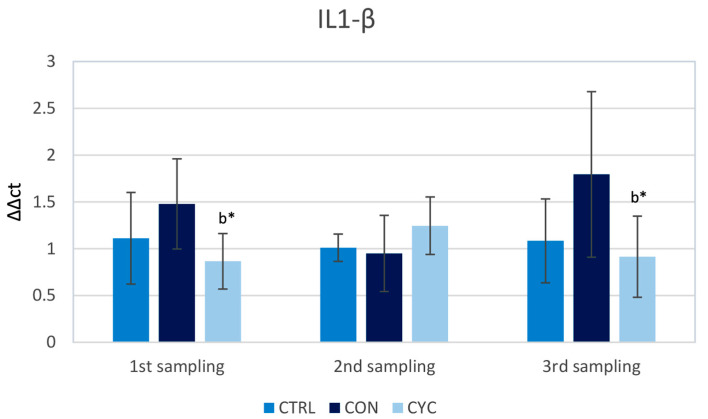
Influence of continuous and cyclic application of probiotic feed on the relative gene expression of cytokine *il1-β* in the trout gut. b—significantly different from group CON; * *p* < 0.05.

**Figure 6 animals-13-01892-f006:**
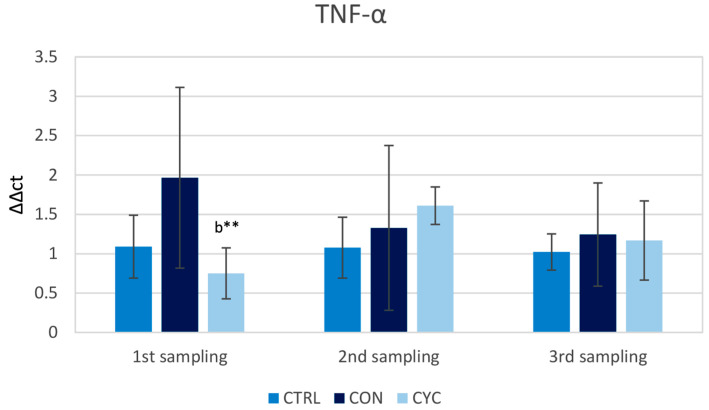
Influence of continuous and cyclic application of probiotic feed on the relative gene expression of cytokine *tnf-α* in the trout gut. b—significantly different from group CON; ** *p* < 0.01.

**Figure 7 animals-13-01892-f007:**
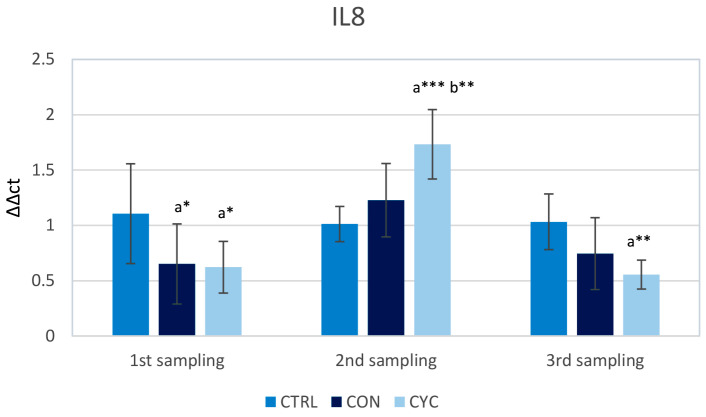
Influence of continuous and cyclic application of probiotic feed on the relative gene expression of cytokine *il8* in the trout gut. a—significantly different from the control group; b—significantly different from group CON; * *p* < 0.05, ** *p* < 0.01, *** *p* < 0.001.

**Figure 8 animals-13-01892-f008:**
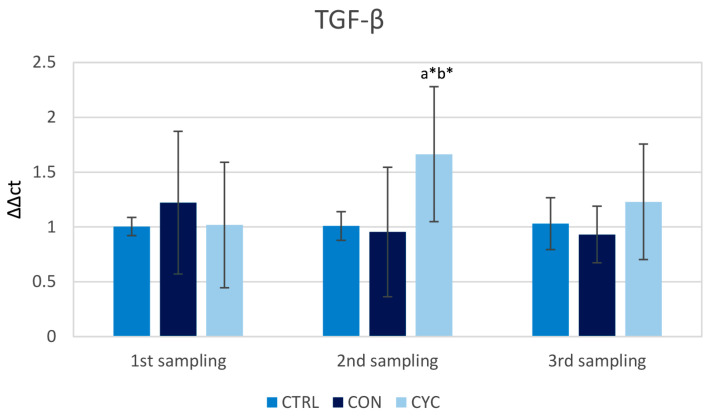
Influence of continuous and cyclic application of probiotic feed on the relative gene expression of cytokine *tgf-β* in the trout gut. a—significantly different from the control group; b—significantly different from group CON; * *p* < 0.05.

**Figure 9 animals-13-01892-f009:**
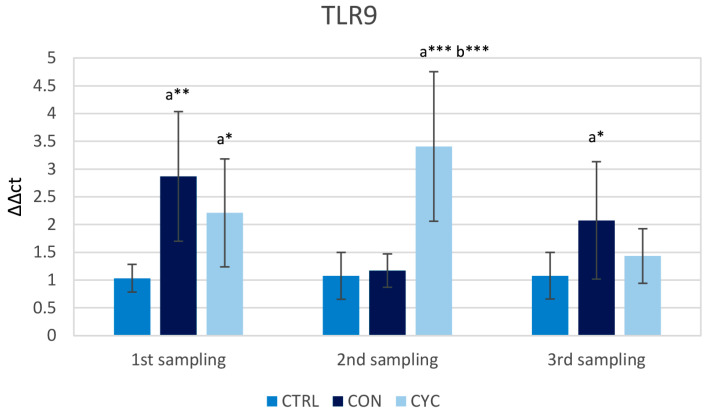
Influence of continuous and cyclic application of probiotic feed on the relative gene expression of *tlr9*. a—significantly different from the control group; b—significantly different from group CON; * *p* < 0.05, ** *p* < 0.01, *** *p* < 0.001.

**Figure 10 animals-13-01892-f010:**
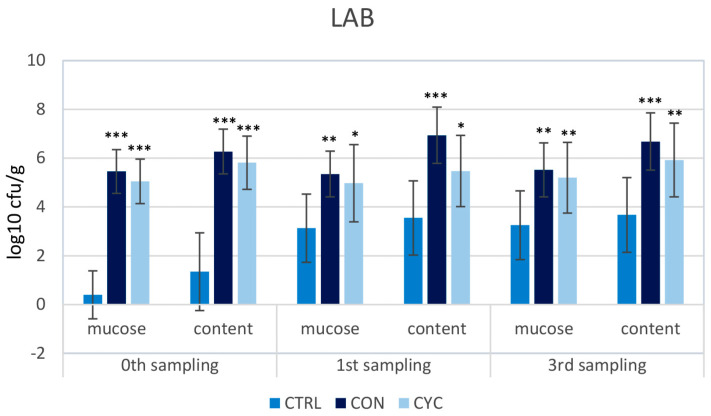
Influence of continuous and cyclic application of probiotic feed on the amounts of lactic acid bacteria adhered to the gut mucosa and in the gut content of rainbow trout. Significant differences from the control group are marked with stars. Levels of significance: * *p* < 0.05; ** *p* < 0.01; *** *p* < 0.001.

**Figure 11 animals-13-01892-f011:**
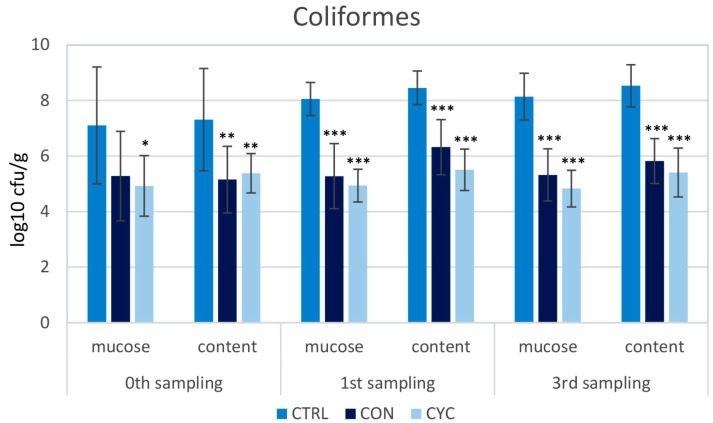
Influence of continuous and cyclic application of probiotic feed on the amounts of coliform bacteria adhered to the gut mucosa and in the gut content of rainbow trout. Significant differences from the control group are marked with stars. Levels of significance: * *p* < 0.05; ** *p* < 0.01; *** *p* < 0.001.

**Figure 12 animals-13-01892-f012:**
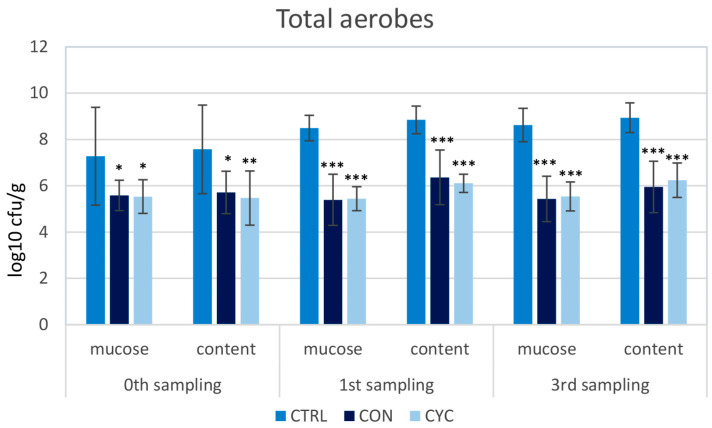
Influence of continuous and cyclic application of probiotic feed on the amounts of total aerobic bacteria adhered to the gut mucosa and in the gut content of rainbow trout. Significant differences from the control group are marked with stars. Levels of significance: * *p* < 0.05; ** *p* < 0.01; *** *p* < 0.001.

**Figure 13 animals-13-01892-f013:**
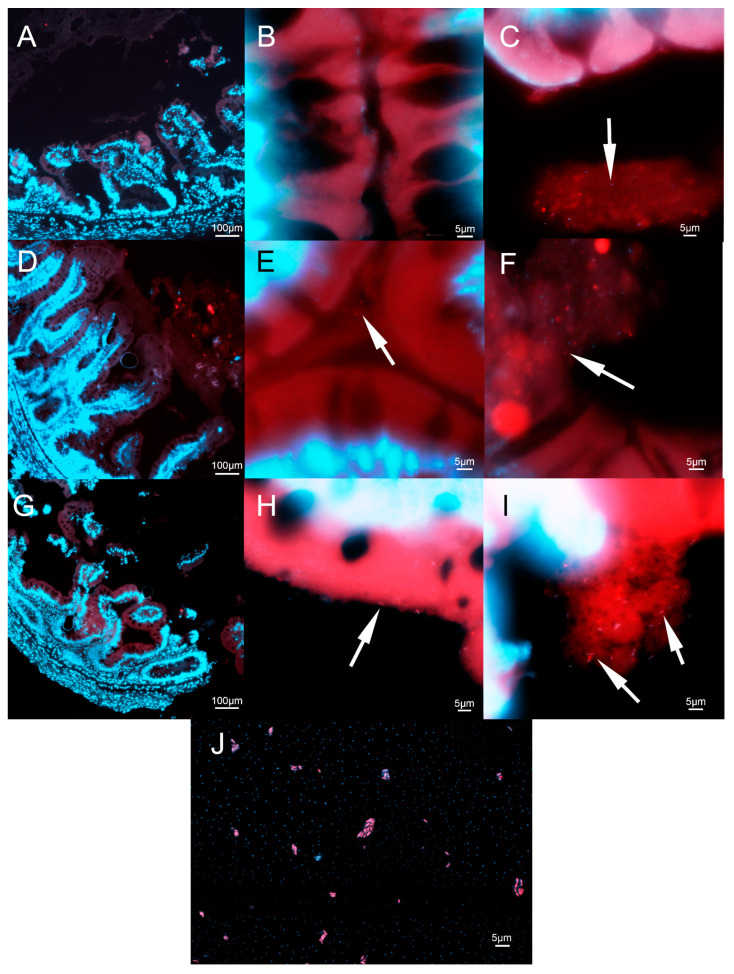
The intestine of rainbow trout after staining. The intestinal section of the control (**A**–**C**), continually fed (**D**–**F**), and cyclically fed groups (**G**–**I**). Bacterial smear of *L. plantarum* for confirmation of the function of staining (**J**). Arrows indicate *L. plantarum*.

**Figure 14 animals-13-01892-f014:**
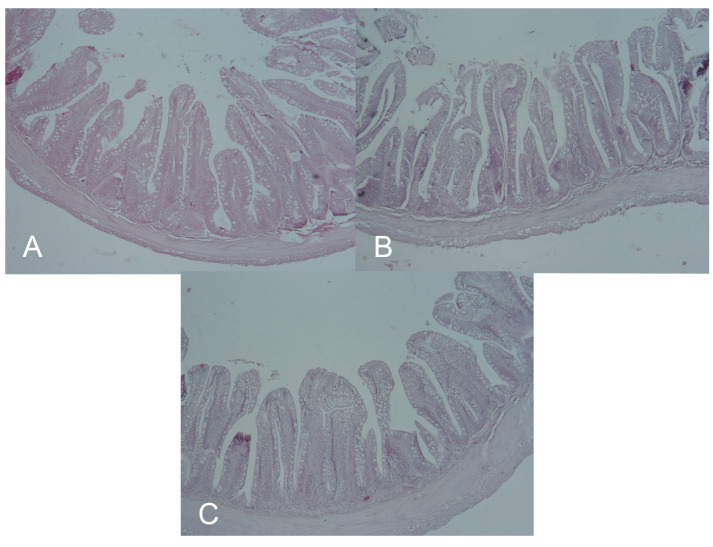
Histological sections of rainbow trout intestine from the CTRL (**A**), CYC (**B**) and CON (**C**) groups.

**Figure 15 animals-13-01892-f015:**
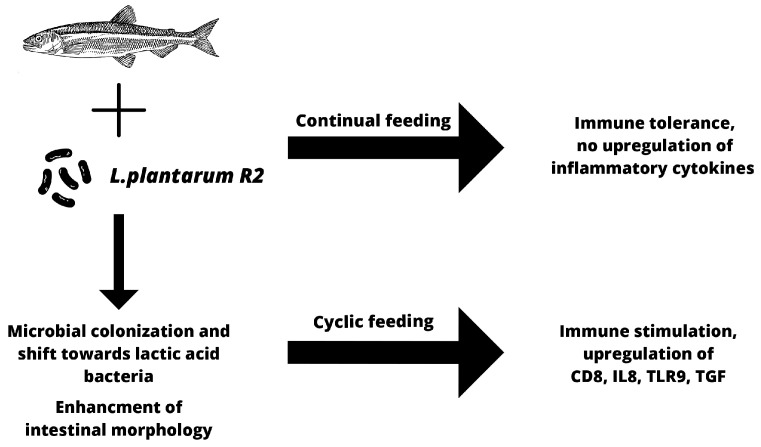
Summarization of effects of probiotic feed using autochthonous bacteria *L. plantarum* R2.

**Table 1 animals-13-01892-t001:** Mass of components for preparation of probiotic aquafeed.

Components	Weight (g)
Pellets (EFICO Enviro 921 3 mm)	1000
25% suspension of *L. plantarum* R2 in sterile saline	120
Colloidal silicon dioxide (Aerosil^®^ 200)	10
Pregelatinized maize starch (Starch 1500^®^)	10

**Table 2 animals-13-01892-t002:** Analytical constituents of control feed (coated with silicon-starch hydrogel) and probiotic feed (containing *L. plantarum* R2 in the silicon-starch hydrogel).

Constituents	Control (Coated)	Probiotic
Dry matter (DM), g/kg	850.7	858.3
Nitrogenous compounds, g/kg	439.8	439.4
Crude fiber, g/kg	21.9	20.3
Fat, g/kg	190.4	192.5
Ash, g/kg	74.1	75.1
Organic matter, g/kg	776.6	793.3
Total calcium, g/kg	10.4	10.4
Total phosphorus, g/kg	5.9	5.6
Magnesium, g/kg	2.2	2.2
Sodium, g/kg	3.5	3.5
Potassium, g/kg	6.2	6.3
Iron, mg/kg	232.9	244.9
Manganese, mg/kg	17.1	20.3
Zinc, mg/kg	85.8	90.7
Copper, mg/kg	70.0	78.4
Aspartic acid, g/kg DM	51.3	50.7
Threonine, g/kg DM	24.6	24.3
Serine, g/kg DM	25.8	25.9
Glutamic acid, g/kg DM	93.5	89.3
Proline, g/kg DM	26.7	25.4
Glycine, g/kg DM	27.8	26.6
Alanine, g/kg DM	30.9	28.6
Valine, g/kg DM	21.3	24.0
Isoleucine, g/kg DM	14.7	16.5
Leucine, g/kg DM	42.8	42.9
Tyrosine, g/kg DM	14.4	15.3
Phenylalanine, g/kg DM	23.7	24.4
Histidine, g/kg DM	19.8	18.7
Lysine, g/kg DM	35.9	36.4
Arginine, g/kg DM	30.8	31.1
Methionine, g/kg DM	10.5	9.5
Cystine, g/kg DM	5.9	5.7

**Table 3 animals-13-01892-t003:** Sequences of primers used in qPCR and relevant references containing qPCR conditions.

Gene	Primer Sequence	Reference
*β-actin* F	GGACTTTGAGCAGGAGATGG	[23]
*β-actin* R	ATGATGGAGTTGTAGGTGGTCT	
*igm* F	ACCTTAACCAGCCGAAAG	[24]
*igm* R	TGTCCCATTGCTCCAGTC	
*cd4* F	CCTGCTCATCCACAGCCT	[24]
*cd4* R	CTTCTCCTGGCTGTCTGA	
*cd8* F	AGTCGTGCAAAGTGGGA	[24]
*cd8* R	GGTTGCAATGGCATACAG	
*il1-β* F	ACATTGCCAACCTCATCATCG	[25]
*il1-β* R	TTGAGCAGGTCCTTGTCCTTG	
*tnf-α* F	GGGGACAAACTGTGGACTGA	[26]
*tnf-α* R	GAAGTTCTTGCCCTGCTCTG	
*il8* F	CACAGACAGAGAAGGAAGGAAAG	[23]
*il8* R	TGCTCATCTTGGGGTTACAGA	
*tgf-β* F	TCTGAATGAGTGGCTGCAAG	[27]
*tgf-β* R	GGTTTCCCACAATCACAAGG	
*tlr9* F	GCAACCAGTCCTTCCACATT	[28]
*tlr9* R	AAACCCAGGGTAAGGGTTTG	

F—forward, R—reverse.

**Table 4 animals-13-01892-t004:** Morphometric parameters of the intestine of rainbow trout fed probiotics continually (CON) and cyclically (CYC) and the control (CTRL).

	CTRL	CON	CYC
Average cross-section surface (µm^2^)	368,505.4 ± 235,495.5	533,749.5 ± 218,978.6 a ***	706,911.7 ± 267,860.5 a *** b *
Average villus height (µm)	1404.6 ± 289.3	1374.8 ± 296.4	1512.7 ± 396.6 b *
Average villus width (µm)	460.9 ± 92.0	432.6307 ± 101.9	435.8 ± 93.2

a—a significant difference to CTRL, b—a significant difference to CON. Levels of significance: * *p* < 0.05; *** *p* < 0.001.

## Data Availability

The data presented in this study are available on request from the corresponding author.

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
