# Peer review of "Feeding-Regime-Dependent Intestinal Response of Rainbow Trout after Administration of a Novel Probiotic Feed"

_animals, 2023, doi:10.3390/ani13121892_

Round 1
Reviewer 1 Report
The present study compares a continuous application of a L. plantarum probiotic feed to a cyclic application, and evaluates expression of immune-related genes and gut bacteria content. Probiotic feeds have a lot of potential in aquaculture applications, so this work is relevant and timely. The approach is technically sound and adequate for answering the questions posed. There are however, several issues with the presentation, interpretation, and discussion of the results that need to be addressed. Throughout the discussion, results are consistently misrepresented. Frequently, the authors state trends or observed differences that are not present in their results or supported by their statistical analysis. Furthermore, the discussion does not adequately explore why the results they observed may have occurred. For many of the results, they are simply re-stated with a statement that they are similar to other studies (with little further discussion as to why). Therefore, I suggest a careful re-write of the discussion that carefully addresses the observed results reported in the manuscript. This will also require a softening of the reported confidence about the immune response results. There are also many places where grammar or word choice can be improved. These are largely cosmetic changes, but some are necessary to make the text readable. Specific comments are detailed below.
Line Comments:
Line 22: Perhaps replace “… which is associated with high energy demands.” With “… which can lead to unwanted increases in energy demand.”
Line 27: Replace “resistance” with “immune function”.
Line 28: Add a comma after “developed”.
Line 32: Replace “On the other hand,” with “In the cyclic treatment”.
Line 36: Replace “it” with “, this probiotic feed”.
Line 47: Add a period after “limited” and start a new sentence with “Therefore”.
Line 50: Either put “Therefore” at the beginning of the sentence, or remove it.
Line 53: Start a new sentence with “With”.
Line 72: After the comma, rephrase to “, and include other beneficial effects such as nutritional competition and immunomodulation [10].”
Line 83: Replace “as an” with “by an observed”, and remove “the” before “metabolic activity”.
Lines 88-89: What is meant by “very appropriate and perspective”? Perhaps replace with “a potentially valuable tool going forward” or something to that effect.
Line 93: Add “of LAB” after “Both Strains”.
Line 94-96: It may be worth adding a brief description of what the antibiotic susceptibility requirements are for readers who are not familiar.
Line 101: Replace “because” with “as”.
Line 114: Replace “to” with “in”.
Line 185: Replace “thus it” with “and”.
Line 190: It is challenging to understand when exactly the samples were taken from just reading the text. It would be helpful to have a figure showing a timeline of feeding treatments and when samples were taken.
Line 277: Start a new sentence after “test”.
Results: Results are well presented and straightforward.
All Figures: It may improve readability to change the figure text to black. The grey text is going to be challenging to read on many screens.
Figure 12: Because the label for Figure 12I is white, it is impossible to see in this figure. Consider making this label black.
Line 327: The caption states that “a” indicates significantly different than the control group, but both labels are “b”. I see the point of keeping these labels consistent (as this scheme is used throughout), but I suggest changing this to be similar to figure 1 where only the “b” label is described in the figure caption.
Line 436: It appears a default/placeholder table caption has been left in place.
Line 444: “probability of their good adaptation” I’m not entirely sure what is meant here.
Line 456-459: Given that feed acceptance is very important, it may be worth including these results rather than simply mentioning them offhand.
Line 460-461: Replace “It is known that” with “Often, “.
Line 465: Perhaps find a better term to use than “weaker and risky farms”.
Results: Be sure to reference figures when discussing relevant results in the discussion.
Line 474: I don’t necessarily agree with your interpretation of the results here. You state “Our results show that relative gene expression is elevated in fish after the reintroduction of probiotic bacteria into the intestinal tract, but not after continual feeding.”. In your results, IgM is not significantly elevated relative to the 1st sampling and is not significantly elevated relative to the control. It is only elevated relative to the continuous application, and only at sample point 2 (unless I’m reading the figure incorrectly). Therefore, you could say that IgM expression was significantly higher in the cyclical treatment than in the constant treatment, but not significantly higher than the control. This is a big difference.
Line 520-521: I’m not sure this thoroughly describes your observed results. You observed reduced expression of IL1 in the cyclical treatment at two time points, yet it is only mentioned that IL1 expression was not increased (on line 526). The significant reduction in IL1 expression should be explicitly discussed.
Line 531-534: This is fine, but how does it relate to the observed IL1 results presented in this paragraph?
Paragraph beginning on line 535: In this paragraph, it is stated that TNF-a was increased over the control in your experimental treatments. Your results do not support this statement. In figure 5, there are no reported significant increases between your experimental treatments and controls at any sample point. The only observed difference was between the cyclical and constant treatment in the 1st sample. This paragraph misrepresents your observed results in its attempts to put them in context with the literature.
Paragraph beginning on line 546: While the CYC treatment is discussed thoroughly, the CON treatment is not discussed or explained.
Line 562: Replace “proven” with “observed”
Paragraph beginning on line 559: Again, the CON treatment is not discussed. Also, while it is stated that there was increased TGF-b expression in the cyclical treatment at sample 2, potential causes or mechanisms are not mentioned.
Line 579: Replace “proven” with “suggested”
Line 581: Replace “experimental” with “CYC”
Line 582: It is stated that TLR9 expression is elevated in the experimental groups during the entire statement, but your results do not support this. The CON treatment was not upregulated in the second sample and the CYC treatment was not upregulated in the third sample.
Paragraph beginning on line 586: This paragraph does a great job of presenting the results, discussing their significance, and placing them in the broader context of the literature. Nice work.
Line 619: Replace “proving” with “suggesting” as proving is too strong of a statement.
Line 636: Replace “aquacultures” with “aquaculture operations”.
Line 636: Why would a continuous application be better suited for lower water quality/feed quality? I’m not sure you can make this conclusion without explicitly testing it in your study.
There are many places where grammar or word choice can be improved. These are largely cosmetic changes, but some are necessary to make the text readable.
Author Response
Answers to Reviewer 1:
In the first place, we would like to express our thanks for checking the manuscript and giving stimulating suggestions. We tried to correct the manuscript based on your demands and/or explain some points.
Line Comments:
Line 22: Perhaps replace “… which is associated with high energy demands.” With “… which can lead to unwanted increases in energy demand.”
The sentence has been reformulated.
Line 27: Replace “resistance” with “immune function”.
The words have been changed.
Line 28: Add a comma after “developed”.
The comma was added.
Line 32: Replace “On the other hand,” with “In the cyclic treatment”.
Line 36: Replace “it” with “, this probiotic feed”.
Lines 32, 36: The expressions have been changed.
Line 47: Add a period after “limited” and start a new sentence with “Therefore”.
Line 50: Either put “Therefore” at the beginning of the sentence, or remove it.
Line 53: Start a new sentence with “With”.
Line 72: After the comma, rephrase to “, and include other beneficial effects such as nutritional competition and immunomodulation [10].”
Lines 47, 50, 53, 72: The sentences have been modified.
Line 83: Replace “as an” with “by an observed”, and remove “the” before “metabolic activity”.
The sentence was corrected.
Lines 88-89: What is meant by “very appropriate and perspective”? Perhaps replace with “a potentially valuable tool going forward” or something to that effect.
Line 93: Add “of LAB” after “Both Strains”.
Lines 88-89, 93: The expressions have been changed.
Line 94-96: It may be worth adding a brief description of what the antibiotic susceptibility requirements are for readers who are not familiar.
The information concerning the antimicrobial susceptibility of both strains on antibiotics listed by EFSA has been added to the text.
Line 101: Replace “because” with “as”.
Line 114: Replace “to” with “in”.
Line 185: Replace “thus it” with “and”.
Lines 101, 114, 185. The words were changed.
Line 190: It is challenging to understand when exactly the samples were taken from just reading the text. It would be helpful to have a figure showing a timeline of feeding treatments and when samples were taken.
The sampling scheme was prepared and inserted into the Material and methods.
Line 277: Start a new sentence after “test”.
The sentence has been corrected.
Results: Results are well presented and straightforward.
The results have been slightly modified based on the demands of another reviewer.
All Figures: It may improve readability to change the figure text to black. The grey text is going to be challenging to read on many screens.
The figure´s text has been changed to black.
Figure 12: Because the label for Figure 12I is white, it is impossible to see in this figure. Consider making this label black.
The color of the image labels has been changed to make the text readable.
Line 327: The caption states that “a” indicates significantly different than the control group, but both labels are “b”. I see the point of keeping these labels consistent (as this scheme is used throughout), but I suggest changing this to be similar to Figure 1 where only the “b” label is described in the figure caption.
In all graphs, the lowercase letter "a" indicates a significant difference from the control group, and "b" is a significant difference from the continuous group (CON). In Figure 4, where no significant differences from the control were noted, the description of the letter "a" was removed.
Line 436: It appears a default/placeholder table caption has been left in place.
The sentence has been removed.
Line 444: “probability of their good adaptation” I’m not entirely sure what is meant here.
The expression was modified for its better understanding, and the idea was to point out that the original (autochthonous) strains are better adapted to the conditions of the organism of the animal species from which they were isolated. Non-autochthonous strains are often used in practice (e.g. for fish from terrestrial animals or of unclear origin) and their effectiveness is often insufficient.
Line 456-459: Given that feed, acceptance is very important, it may be worth including these results rather than simply mentioning them offhand.
This experiment was carried out in cooperation with partners from the Czech Republic (from Mendel University, Veterinary Research Institute, and Veterinary University in Brno), where a wide range of parameters was monitored. In addition to production parameters, also hematological and biochemical. Due to the huge number of results, these were divided and the Czech colleagues prepared a publication focused on these parameters. The manuscript was sent to another journal and now is under review.
Line 460-461: Replace “It is known that” with “Often, “.
Line 465: Perhaps find a better term to use than “weaker and risky farms”.
The expressions have been modified.
Results: Be sure to reference figures when discussing relevant results in the discussion.
Links to the relevant figures have been added to the discussion.
Line 474: I don’t necessarily agree with your interpretation of the results here. You state “Our results show that relative gene expression is elevated in fish after the reintroduction of probiotic bacteria into the intestinal tract, but not after continual feeding.”. In your results, IgM is not significantly elevated relative to the 1st sampling and is not significantly elevated relative to the control. It is only elevated relative to the continuous application, and only at sample point 2 (unless I’m reading the figure incorrectly). Therefore, you could say that IgM expression was significantly higher in the cyclical treatment than in the constant treatment, but not significantly higher than the control. This is a big difference.
The description of the results has been changed.
Line 520-521: I’m not sure this thoroughly describes your observed results. You observed reduced expression of IL1 in the cyclical treatment at two time points, yet it is only mentioned that IL1 expression was not increased (on line 526). The significant reduction in IL1 expression should be explicitly discussed.
The discussion regarding IL1 has been modified.
Line 531-534: This is fine, but how does it relate to the observed IL1 results presented in this paragraph?
This comment has been removed.
Paragraph beginning on line 535: In this paragraph, it is stated that TNF-a was increased over the control in your experimental treatments. Your results do not support this statement. In figure 5, there are no reported significant increases between your experimental treatments and controls at any sample point. The only observed difference was between the cyclical and constant treatment in the 1st sample. This paragraph misrepresents your observed results in its attempts to put them in context with the literature.
The discussion regarding TNF-a has been modified.
Paragraph beginning on line 546: While the CYC treatment is discussed thoroughly, the CON treatment is not discussed or explained.
The discussion has been added.
Line 562: Replace “proven” with “observed”
The expression has been changed.
Paragraph beginning on line 559: Again, the CON treatment is not discussed. Also, while it is stated that there was increased TGF-b expression in the cyclical treatment at sample 2, potential causes or mechanisms are not mentioned.
The discussion has been modified.
Line 579: Replace “proven” with “suggested”
Line 581: Replace “experimental” with “CYC”
Lines 579, 581: The expressions have been changed.
Line 582: It is stated that TLR9 expression is elevated in the experimental groups during the entire statement, but your results do not support this. The CON treatment was not upregulated in the second sample and the CYC treatment was not upregulated in the third sample.
The discussion has been modified.
Paragraph beginning on line 586: This paragraph does a great job of presenting the results, discussing their significance, and placing them in the broader context of the literature. Nice work.
Thank you very much.
Line 619: Replace “proving” with “suggesting” as proving is too strong of a statement.
Line 636: Replace “aquacultures” with “aquaculture operations”.
Lines 619,636: The expressions have been changed.
Line 636: Why would a continuous application be better suited for lower water quality/feed quality? I’m not sure you can make this conclusion without explicitly testing it in your study.
The obtained results show that with long-term application, the immune system adapts to our probiotic bacteria and there is no unnecessary stimulation of it, which is energy-intensive. At the same time, the intestinal microbiota is stabilized for a long time, which contributes to the health of the fish. When breeding salmonids, the exchange/circulation/movement of water is fast, so if it has a lower quality (higher content of pathogens, xenobiotics, etc.), the short-term application is insufficient for long-term maintenance of a healthy intestinal microbiota, since probiotic bacteria do not remain in the water environment, as is the case for example in carp fish.
Reviewer 2 Report
This manuscript entitled “Feeding Regime-Dependent Intestinal Response of Rainbow Trout After Administration of a Novel Probiotic Feed” mainly focused on the impacts of two probiotic feed applications (continuous feeding and intermittent feeding) on gut health by evaluating the structure of intestinal flora and the key factors related to the intestinal immune response in rainbow trout (Oncorhynchus mykiss). It will provide the necessary foundation to better understand the beneficial actions of probiotic bacteria on the host fishes including rainbow trout and develop an optimal scheme for practical application in aquaculture.
In this case, the paper should be carefully proofread throughout for language to eliminate several spelling and grammar errors.
Major comments:
1. In “2.2 Experimental animals and sampling” section (Line 179-180), what feed were used in rainbow trout during the adaptation period? Commercial feed or other feed? Please offer more information.
2. In “2.4 Microbiological screening” section, samples obtained on the 4th, 7th, and 11th week were used for assessing the intestinal microbiota of trout. However, the gene expression analysis in this study was carried out by using intestinal tissues from the 7th, 9th, and 11th week. Why did you choose different intestinal samples for these two measurements?
3.Regarding the name of the genes, it is accepted that teleost genes are represented in lower case and italicized (e.g. β-actin, tlr9). The authors may check what is suggested for gene and protein nomenclature of teleost.
4.The letters annotated in the graphs (Fig.1-Fig.8) did not meet the statistical difference criteria according to Tukey's test. ANOVA followed by Tukey’s comparison was carried out for analysis of differences among 3 or more groups. For example, in Fig.1, the gene expression of IgM protein was significantly higher in the CYC group than that of the CON group. So different letters are annotated in the figures to indicate statistical differences.
Moreover, the figures in this manuscript need to be optimized for the best aesthetic results.
Minor comments:
1.Check the symbols or codes for volume unit in this study according to the information to related guides. For example, in Line 176, please replace “in three 1000 l tanks” with “in three 1000 L tanks”. Also, there were similar errors in the other part of this manuscript.
2. In Line 211, was it “0,5 μl of both primers”? I think there was a spelling error. Please revise it.
3. In Line 227, what was “a sterile physiological solution”? Was it PBS, isotonic saline solution or other solution? Please offer more information.
4. In “3.4 Morphometry of the Intestine” section, a representative picture needs to be given for each group.
5. In “Results” section, there were many sentences with over-stating results. Some statements can be describes in “Discussion” section. For example, Line 389-392. Please reduce the overall length of “Results” section in this manuscript.
6. Line 575: what is PAMP? Please give the full name for this abbreviation.
Therefore, this manuscript will be reconsidered after major revision.

The paper should be carefully proofread throughout for language to eliminate some spelling and grammar errors.
Author Response
Answers to reviewer 2:
In the first place, we would like to express our thanks for checking the manuscript and giving stimulating suggestions. We tried to correct the manuscript based on your demands and/or explain some points.
Major comments:
- In “2.2 Experimental animals and sampling” section (Line 179-180), what feed were used in rainbow trout during the adaptation period? Commercial feed or other feed? Please offer more information.
The information has been added.
- In “2.4 Microbiological screening” section, samples obtained on the 4th, 7th, and 11thweek were used for assessing the intestinal microbiota of trout. However, the gene expression analysis in this study was carried out by using intestinal tissues from the 7th, 9th, and 11th week. Why did you choose different intestinal samples for these two measurements?
At zero sampling, after 4 weeks of application of probiotics in both experimental groups, a microbiological examination was carried out, where the numbers of monitored bacterial groups can be compared between individual samplings. Since we use the assessment of relative (not absolute) gene expression, the comparison between individual samples is only indicative. Furthermore, with the continuous application of probiotics (between weeks 7 and 11), it would be unnecessary to do a microbiological evaluation after 2 weeks (week 9), as healthy fish under controlled conditions will not undergo significant changes in the microbiota in such a short time. This was also confirmed, as we did a microbiological screening of several pieces of fish even in the 9th week, just to be sure. For better readability of the application and subscriptions, we included a graphic application scheme of the experiment in the revised version of the article.
- Regarding the name of the genes, it is accepted that teleost genes are represented in lower case and italicized (e.g.β-actin, tlr9). The authors may check what is suggested for gene and protein nomenclature of teleost.
We checked the nomenclature for genes and proteins and we modified the gene nomenclature throughout the manuscript.
4.The letters annotated in the graphs (Fig.1-Fig.8) did not meet the statistical difference criteria according to Tukey's test. ANOVA followed by Tukey’s comparison was carried out for analysis of differences among 3 or more groups. For example, in Fig.1, the gene expression of the IgM protein was significantly higher in the CYC group than that of the CON group. So different letters are annotated in the figures to indicate statistical differences.
Statistical analysis was done using Tukey's test, where we compared all 3 groups - control, cyclic and continuous. Statistically significant differences are standardly marked, while in the entire manuscript the letter "a" expresses a significant difference compared to the control group and the letter "b" compared to the continuous group. The inclusion of the letter "c" is not necessary, since if the cyclic group differs from the control or continuous group, it is expressed by the letters "a" and "b". The number of stars indicates the significance level at which the groups differ.
Moreover, the figures in this manuscript need to be optimized for the best aesthetic results.
Figures have been graphically modified.
Minor comments:
- Check the symbols or codes for volume unit in this study according to the information to related guides. For example, in Line 176, please replace “in three 1000 l tanks” with “in three 1000 L tanks”. Also, there were similar errors in the other part of this manuscript.
The units have been corrected throughout the manuscript.
- In Line 211, was it “0,5 μl of both primers”? I think there was a spelling error. Please revise it.
- 3. In Line 227, what was “a sterile physiological solution”? Was it PBS, isotonic saline solution, or other solution? Please offer more information.
Lines 211, 227: The expressions have been corrected.
- In “3.4 Morphometry of the Intestine” section, a representative picture needs to be given for each group.
The figures have been added to the manuscript.
- In “Results” section, there were many sentences with over-stating results. Some statements can be describes in “Discussion” section. For example, Line 389-392. Please reduce the overall length of “Results” section in this manuscript.
The “Results” have been modified, but we also had to take into account the requests of other reviewers who asked for additional information.
- Line 575: what is PAMP? Please give the full name for this abbreviation.
The abbreviation has been explained.
Reviewer 3 Report
The study by Ratvaj et al. is an interesting, focused and comprehensive approach to the immunological responses of an important species under the effect of probiotic treatment. It is a well-designed and well-written article, and therefore I have only some suggestions that the authors can find in the attached pdf.

Author Response
Answers to reviewer 3:
In the first place, we would like to express our thanks for checking the manuscript and giving stimulating suggestions. We tried to correct the manuscript based on your demands. All requested changes have been made, including the inclusion of a graphic conclusion in the manuscript.
Reviewer 4 Report
It is a good-quality article, however, some minor faults can be found in it which I marked in the attached MS together with some suggestions to make it even better. I missed some conventional traits of feed and fish performance. Results of microbial screening, described finely in M&M are also missing.

Author Response
Answers to reviewer 4:
In the first place, we would like to express our thanks for checking the manuscript and giving stimulating suggestions. We tried to correct the manuscript based on your demands and/or explain some points.
Most of the requested changes have been performed.
Some points we would like to explain:
In Table 2 we can not change Nitrogenous compounds to „Raw protein“. Based on consultation with the laboratory that performed the feed analysis, nitrogenous substances also include non-protein nitrogen.
Line 225: For better readability of the application and subscriptions, we included a graphic application scheme of the experiment in the revised version of the article.
Lines 236 - 238: Bacterial counts of LAB, enterobacteria, and total aerobes in Figures 10, 11, and 12 (axis “y”) are expressed in log10 cfu/g.
Lines 282, 456-458: This experiment was carried out in cooperation with partners from the Czech Republic (from Mendel University, Veterinary Research Institute, and Veterinary University in Brno), where a wide range of parameters was monitored. In addition to production parameters, also hematological and biochemical. Due to the huge number of results, these were divided and the Czech colleagues prepared a publication focused on these parameters. The manuscript was sent to another journal and now is under review. We are very sorry that it is not already published and we cannot refer to it.
Round 2
Reviewer 1 Report
The authors have taken some good steps towards improving their paper. I particularly like the addition of figure 1 which makes the study design and sampling points much clearer. However, significant problems remain. The largest issue is that the conclusions made in this study largely lack support from the results. At many points, results that do not support the narrative (that the CYC treatment caused immunostimulation and the CON treatment caused the immune system to adapt to the probiotic) were ignored or arbitrarily dismissed. Furthermore, results that showed mixed or ambiguous support for the conclusions were stated as though they were strong evidence. The result is that a study with somewhat ambiguous results has been presented as though there is clear evidence of the stated conclusions.
What is needed is an a priori set of hypotheses and a detailed description of what conditions need to be met to support those hypotheses. For example, if the authors hypothesize that a cyclical application of the probiotic feed will result in immunostimulation, then there should be an explicit set of conditions that must be met for that hypothesis to be supported. Would all cytokines need to be upregulated at all sample points to support this? What does it mean if some are upregulated, and some are not. What does it mean if they are upregulated at some time points and not others. Lay this out explicitly and thoroughly explore your results in the context of your hypotheses. Currently, it appears that a conclusion was reached, and then the data was interpreted in a way that most supported that conclusion which is not acceptable.
Fixing these issues will require a considerable re-write to the discussion and will somewhat change the overall story, but this is a necessary step for this manuscript.
Line Comments:
Line 49: Replace “following” with “coming”.
Line 55 – 56: Finish this sentence with something to the effect of “[…] there has been a need to find alternative solutions that serve a similar role.”
Line 61: Replace “have been” with “are”
Line 295: The threshold for significance should be stated in this section (e.g. “Differences between groups were considered significant with an alpha of […]”). Also, the full results (d.f., F value, p-value) of ANOVA and Tukey’s Tests should be reported throughout, not just the p-values of the Tukey’s tests.
Line 319: starting at “but its level […]” replace with “but there appeared to be a trend of increasing expression in the CON and CYC treatments respectively (Figure 3).”
Line 328-334: The CON group did not have significantly higher relative expression of cd8 in the first sample, and unlike for cd4, the CYC treatment’s expression was lower than the control. In the second sampling, the CYC treatment has significantly higher expression of cd8 where that was not the case for cd4. The trend in the second sample for cd4 is not present for cd8. Therefore, I don’t think it is accurate to state that “The trend of the relative gene expression level for cd8 was very similar to that for the cd4”. This statement should be removed.
Line 345-347: I’m not sure where this increasing trend is in the CON group for il1-b. It does not seem to increase with each successive sampling, and it has a lower value than the control in the second sampling.
Line 350: Add “but were not significantly different than the control.”
Line 362: Add “but was not significantly different from the control.”
Line 414: Add “but the CYC group was not significantly different from the control.”
Line 516: Remove “to them”.
Line 530: It is stated that IgM expression is elevated in the CYC group relative to the CON group after reintroduction of probiotic bacteria, but it is important to note that it was only elevated in one sampling. There should be some discussion as to why it may only be temporarily elevated relative to the constant treatment, and the temporary nature of the IgM elevation should also be explicitly noted in the discussion. Currently it is written as if it remains significantly higher than the CON group in all of the sampling events. Additionally, the fact that it is not significantly upregulated relative to the control is critical. If expression is not significantly higher than the control, the narrative that the CYC treatment is stimulating the intestinal immune system is not supported by IgM.
Line 559: I think the cd4 results are well explained in this paragraph, but it is important to not that the narrative that the CYC treatment is stimulating the intestinal immune system is also not supported by cd4 as there is not a significant increase in the CYC treatment relative to the control and CON. Furthermore, they provide support for immunostimulation in the CON treatment as there was significant upregulation of CD4 in the first sampling. This is contrary to the conclusions made in the paper.
Line 569: It is stated “As observed in our experiment, the relative gene expression for this molecule was highly elevated after the fish in the cyclic group started receiving the probiotic feed again, (Figure 4), indicating the immunomodulatory potential of this feeding regime.” While there was upregulation of cd8 in the CYC treatment for the second sampling, it was downregulated in the first, and not significantly different from the CTRL or CON treatments in the 3rd sampling. Therefore, I think it is really challenging to make the argument that the cyclic treatment is creating this immunostimulant effect with any confidence, especially when IgM, cd4, TNF-a, IL1-b, IL8, etc. do not support this.
Lines 584-594: I would argue that your results here are not supporting your claim that the CYC treatment has an immunostimulant effect.
Line 590-593: If you are making the claim that the CON treatment results in immunostimulation in the first sample, why do you later conclude that the CON treatment does not cause an increased immune response? Also, there was not a significant difference in CD4 expression in the CON treatment as is stated here.
Line 602-622: Here again, the TNF-a results do not support the claim that the CYC treatment results in immunostimulation.
Line 634-642: The statement here that the CON treatment results are evidence of adaptation to the presence of beneficial bacteria is possibly true, but given the other cytokine responses it is a difficult argument to make with any confidence. Ultimately, instead of treating each cytokine separately, the cumulative response should be used to make the determination of what is happening. If the same trend observed here with IL8 were mirrored with other cytokines, then there is a good case. The TLR9 results certainly are not consistent with the idea that the CON treatment results in adaptation to the presence of beneficial bacteria for example.
Line 657: Based on the two examples from the literature you provided, you would expect to see either an increase or decrease in TGF-b expression after treatment with a probiotic feed. Given these examples, it is difficult to understand what your expected result would be for your study. In the CON treatment, you observed neither an increase nor a decrease relative to the control and state that this is evidence that the diet is safe. Then, you state that the increase observed in the CYC group indicates that it is an effective treatment for intestinal inflammation. Which is it? Does that mean that the CON treatment is not an effective treatment for intestinal inflammation, or does it mean that the CYC group is unsafe?
Lines 684-696: This is a good discussion of TLR9 expression, but it may be valuable to discuss why there was no observed upregulation for the CON treatment in the second sampling and the CYC treatment in the third sampling. How should that be interpreted?
Lines 740-743: Regarding the conclusion that the continuous application results in no significant stimulation of the intestinal immune response, you will need to modify this. There was significant upregulation of TLR9 and CD4 at various timepoints. While the other tested cytokines were not upregulated, it is important not to ignore the ones that were. At the very minimum, this statement should be softened to say that there was stimulation of some cytokines, but not others. This paints a very different picture than what is concluded currently.
Lines 743-747: There was not significant stimulation of igm over the control. It was only upregulated relative to the CON treatment. Also, the actual results are less clear-cut than what is being presented here. For cd8, tgf-b, and il8, there was only significant upregulation at one time point. So ultimately there was one instance of observed upregulation of 3 cytokines (at 1 of 3 time points) and no observed upregulation of 3 cytokines (at any time point). Does this constitute clear evidence of immunostimulation? The case that it does largely hinges on ignoring the results that do not indicate immunostimulation and overstating the results that maybe show a transitory/temporary mild immunostimulation.
Line 750: This study did not test the efficacy of these two applications of probiotic feed for low water or feed quality. Therefore, how can it be claimed that one of these treatments is more suited to those conditions? While the reasoning for this statement was stated in the response to comments, it doesn’t change that there are no results from this study that can support this claim.
Figure 15: For the continual feeding treatment, what would have constituted a negative change in relative gene expression? It appears that upregulation is interpreted as “immune stimulation” and is not treated negatively. Therefore would it be downregulation? Also, this figure states that there was upregulation of cd8, igM, il8, TLR9, and TGF-b. In fact, there was not observed upregulation in igM. It was not significantly upregulated relative to the control.
There are a few places where wording can be improved, but nothing major. See the line comments.
Author Response
Dear reviewer,
once again, we would like to express our thanks for the revision of the manuscript and additional stimulating comments. Again, we tried to edit the manuscript in accordance with your comments or clarify some points. We introduced hypotheses into the discussion, on the basis of which we reached conclusions. We tried to adjust the discussion according to your requirements. We modified and refined the conclusions of the work. We also corrected the results that were shortened based on the request of another reviewer and some errors occurred when modifying the text.
Answers to line Comments:
Line 49: Replace “following” with “coming”.
The words have been replaced.
Line 55 – 56: Finish this sentence with something to the effect of “[…] there has been a need to find alternative solutions that serve a similar role.”
The sentence has been modified.
Line 61: Replace “have been” with “are”
The expression has been replaced.
Line 295: The threshold for significance should be stated in this section (e.g. “Differences between groups were considered significant with an alpha of […]”). Also, the full results (d.f., F value, p-value) of ANOVA and Tukey’s Tests should be reported throughout, not just the p-values of the Tukey’s tests.
The threshold for P-value has been added to the Materials and methods (subchapter 2.7 Statistical analysis).
Regarding your request to present the complete results of the statistical analysis: Since the data were analyzed for each individual parameter in each sampling, it would be necessary to list all the statistical parameters you requested for each sampling, which would not be clear for readers and it is not common for this type of results. If you insist on providing all the statistical data, we can insert the complete results of the statistical analysis from the statistical program as a supplementary file. This is 45 pages of results including statistical analysis of microbiota and morphometry.
Line 319: starting at “but its level […]” replace with “but there appeared to be a trend of increasing expression in the CON and CYC treatments respectively (Figure 3).”
The expression has been replaced.
Line 328-334: The CON group did not have significantly higher relative expression of cd8 in the first sample, and unlike for cd4, the CYC treatment’s expression was lower than the control. In the second sampling, the CYC treatment has significantly higher expression of cd8 where that was not the case for cd4. The trend in the second sample for cd4 is not present for cd8. Therefore, I don’t think it is accurate to state that “The trend of the relative gene expression level for cd8 was very similar to that for the cd4”. This statement should be removed.
We are sorry – it was mistake, the expression has been removed.
Line 345-347: I’m not sure where this increasing trend is in the CON group for il1-b. It does not seem to increase with each successive sampling, and it has a lower value than the control in the second sampling.
This was also mistake made during the previous revision of the article. Based on the request of another reviewer, we tried to shorten the description of the results and a bad interpretation occurred. The expression was corrected.
Line 350: Add “but were not significantly different than the control.”
Line 362: Add “but was not significantly different from the control.”
Line 414: Add “but the CYC group was not significantly different from the control.”
Lines 350, 362, 414: The comparison with the control group was completed.
Line 516: Remove “to them”.
The sentence has been corrected.
Line 530: It is stated that IgM expression is elevated in the CYC group relative to the CON group after reintroduction of probiotic bacteria, but it is important to note that it was only elevated in one sampling.
There should be some discussion as to why it may only be temporarily elevated relative to the constant treatment, and the temporary nature of the IgM elevation should also be explicitly noted in the discussion. Currently it is written as if it remains significantly higher than the CON group in all of the sampling events. Additionally, the fact that it is not significantly upregulated relative to the control is critical. If expression is not significantly higher than the control, the narrative that the CYC treatment is stimulating the intestinal immune system is not supported by IgM.
The discussion has been modified and extended.
Line 559: I think the cd4 results are well explained in this paragraph, but it is important to not that the narrative that the CYC treatment is stimulating the intestinal immune system is also not supported by cd4 as there is not a significant increase in the CYC treatment relative to the control and CON. Furthermore, they provide support for immunostimulation in the CON treatment as there was significant upregulation of CD4 in the first sampling. This is contrary to the conclusions made in the paper.
The discussion has been modified and extended.
Line 569: It is stated “As observed in our experiment, the relative gene expression for this molecule was highly elevated after the fish in the cyclic group started receiving the probiotic feed again, (Figure 4), indicating the immunomodulatory potential of this feeding regime.” While there was upregulation of cd8 in the CYC treatment for the second sampling, it was downregulated in the first, and not significantly different from the CTRL or CON treatments in the 3rd sampling. Therefore, I think it is really challenging to make the argument that the cyclic treatment is creating this immunostimulant effect with any confidence, especially when IgM, cd4, TNF-a, IL1-b, IL8, etc. do not support this.
The discussion has been edited and the explanation that we expect immunostimulation in the CYC group only in the second sampling - that is, after reintroduction of the probiotic feed - was added.
Lines 584-594: I would argue that your results here are not supporting your claim that the CYC treatment has an immunostimulant effect.
The discussion has been modified and a statement was added that we did not expect an increase in pro-inflammatory cytokines, which is unnecessary in healthy fish.
Line 590-593: If you are making the claim that the CON treatment results in immunostimulation in the first sample, why do you later conclude that the CON treatment does not cause an increased immune response? Also, there was not a significant difference in CD4 expression in the CON treatment as is stated here.
Line 602-622: Here again, the TNF-a results do not support the claim that the CYC treatment results in immunostimulation.
The discussion has been modified in the similar way as for IL1.
Line 634-642: The statement here that the CON treatment results are evidence of adaptation to the presence of beneficial bacteria is possibly true, but given the other cytokine responses it is a difficult argument to make with any confidence. Ultimately, instead of treating each cytokine separately, the cumulative response should be used to make the determination of what is happening. If the same trend observed here with IL8 were mirrored with other cytokines, then there is a good case. The TLR9 results certainly are not consistent with the idea that the CON treatment results in adaptation to the presence of beneficial bacteria for example.
The discussion has been modified.
Line 657: Based on the two examples from the literature you provided, you would expect to see either an increase or decrease in TGF-b expression after treatment with a probiotic feed. Given these examples, it is difficult to understand what your expected result would be for your study. In the CON treatment, you observed neither an increase nor a decrease relative to the control and state that this is evidence that the diet is safe. Then, you state that the increase observed in the CYC group indicates that it is an effective treatment for intestinal inflammation. Which is it? Does that mean that the CON treatment is not an effective treatment for intestinal inflammation, or does it mean that the CYC group is unsafe?
By using references with conflicting results, we wanted to point out a clearly strain-dependent effect on TGF-b gene expression. We have modified discussion towards the anti-inflammatory potential of the probiotic feed.
Lines 684-696: This is a good discussion of TLR9 expression, but it may be valuable to discuss why there was no observed upregulation for the CON treatment in the second sampling and the CYC treatment in the third sampling. How should that be interpreted?
The explanantion has been added.
Lines 740-743: Regarding the conclusion that the continuous application results in no significant stimulation of the intestinal immune response, you will need to modify this. There was significant upregulation of TLR9 and CD4 at various timepoints. While the other tested cytokines were not upregulated, it is important not to ignore the ones that were. At the very minimum, this statement should be softened to say that there was stimulation of some cytokines, but not others. This paints a very different picture than what is concluded currently.
Lines 743-747: There was not significant stimulation of igm over the control. It was only upregulated relative to the CON treatment. Also, the actual results are less clear-cut than what is being presented here. For cd8, tgf-b, and il8, there was only significant upregulation at one time point. So ultimately there was one instance of observed upregulation of 3 cytokines (at 1 of 3 time points) and no observed upregulation of 3 cytokines (at any time point). Does this constitute clear evidence of immunostimulation? The case that it does largely hinges on ignoring the results that do not indicate immunostimulation and overstating the results that maybe show a transitory/temporary mild immunostimulation.
Line 750: This study did not test the efficacy of these two applications of probiotic feed for low water or feed quality. Therefore, how can it be claimed that one of these treatments is more suited to those conditions? While the reasoning for this statement was stated in the response to comments, it doesn’t change that there are no results from this study that can support this claim.
The discussion has been modified in accordance with the results.
Figure 15: For the continual feeding treatment, what would have constituted a negative change in relative gene expression? It appears that upregulation is interpreted as “immune stimulation” and is not treated negatively. Therefore would it be downregulation? Also, this figure states that there was upregulation of cd8, igM, il8, TLR9, and TGF-b. In fact, there was not observed upregulation in igM. It was not significantly upregulated relative to the control.
Figure 15 has been edited in accordance with modified results and your demands.
Reviewer 2 Report
This manuscript entitled “Feeding Regime-Dependent Intestinal Response of Rainbow Trout After Administration of a Novel Probiotic Feed” explored the effect of two probiotic feed applications (continuous feeding and intermittent feeding) on the intestinal immune response and microbiota of rainbow trout (Oncorhynchus mykiss). It provided the technical support for developing and optimizing cost-effective, safe, environment-friendly aquafeed, as well as the effective feeding strategy of probiotic bacteria in aquaculture.
The authors revised the manuscript according to the Reviewers’ comments. But some figures (Fig.2-Fig.12) can be further presented exceedingly clearly and aesthetically.
So, I suggest that this manuscript can be published after revising the manuscript.
Minor comments:
Fig.2-Fig.12 in this manuscript need to be further esthetically presented.
Therefore, this manuscript will be reconsidered after minor revision.
The spelling and grammar errors have been revised.
Author Response
Dear reviewer,
once again, we would like to express our thanks for the revision of the manuscript.
Minor comments:
Fig.2-Fig.12 in this manuscript need to be further esthetically presented.
We have increased the resolution of the images, but we do not know what is meant by the term "esthetically presented", because we used the common type of graphs that are also used in the latest articles of the journal "Animals". If you require further changes, please give us specific instructions.
Round 3
Reviewer 1 Report
The authors have taken some good steps towards improving their paper. Many of the cases of inaccurate descriptions of the results have been addressed. The addition of explicit hypotheses on lines 529-537 are also an important change. There are a handful of remaining grammatical changes that should be made to improve clarity, but these should be caught in copy editing. Also, the inclusion of the test statistic value and degrees of freedom should be included. However, given that the journal guidelines do not explicitly require this, I will leave this up to the editor.
There are a handful of places that need minor re-wording to improve comprehension. This should be handled in copy editing.
Author Response
Dear reviewer,
we let an English teacher check the text of the manuscript. Corrections of grammatical errors are indicated in the text. We have already contacted the editor about the addition of extension statistics, who did not request their insertion in the manuscript. Once again, we thank you for your suggestive advice and comments, and thus for improving the quality of the manuscript.